# Differentially private and decentralized randomized power method

## Abstract

The randomized power method has gained significant interest due to its simplicity and efficient handling of large-scale spectral analysis and recommendation tasks. However, its application to large datasets containing personal user information (e.g., web interactions, search history, personal tastes) raises critical privacy problems. This paper addresses these issues by proposing enhanced privacy-preserving variants of the method. First, we propose a variant that reduces the variance of the noise required in current techniques to achieve Differential Privacy (DP). More precisely, we modify the algorithm and privacy analysis so that the Gaussian noise variance no longer grows linearly with the target rank, achieving the same $(\varepsilon, \delta)$-DP guarantees with lower noise variance. Second, we adapt our method to a decentralized framework in which data is distributed among multiple user devices, strengthening privacy guarantees with no accuracy penalty and a low computational and communication overhead. Our results also include the provision of tighter convergence bounds for both the centralized and decentralized versions, and an empirical comparison with previous work using real recommendation datasets.

## 1 Introduction

The randomized power method has emerged as an efficient and scalable tool for addressing large-scale linear algebra problems central to modern machine learning pipelines (Halko et al., 2011). By constructing an orthonormal basis for a matrix's range in near-linear time, the method scales seamlessly to practical large datasets. Its reliance on simple matrix products ensures compatibility with sparse data representations and enables efficient parallelization and hardware acceleration on modern GPUs and distributed architectures.

Beyond its simplicity, the method provides strong approximation guarantees and accelerates a wide spectrum of applications. It has been used for principal component analysis (PCA) (Journée et al., 2010), singular value decomposition (SVD) (Halko et al., 2011), truncated eigendecompositions (Yuan and Zhang, 2013), and matrix completion (Feng et al., 2018). Extensions have powered recommender systems (e.g., Twitter (Gupta et al., 2013), GF-CF (Shen et al., 2021), BSPM (Shen et al., 2021)), PageRank-style ranking (Ipsen and Wills, 2005), partial differential equations (PDE) solvers (Greengard and Rokhlin, 1997), or large-scale least-squares and linear-system solvers (Rokhlin and Tygert, 2008).

These methods are integrated into large-scale systems involving sensitive user data, whose protection is paramount. Unfortunately, the standard randomized power method does not inherently provide privacy guarantees. While its output might seem less sensitive than the input data matrix, there is no formal guarantee against inference of private information embedded in the outputs of the randomized power method.

To address and quantify privacy leakages, Differential Privacy (DP) has emerged as a powerful framework that provides strong guarantees and mitigates potential privacy leaks of an algorithm, ensuring that the output of an algorithm reveals little about any individual record in the input. Several works have attempted to apply DP to the randomized power method. For example, Hardt and Price (2014); Balcan et al. (2016) developed centralized Differentially Private variants of the power method, whereas Wang and Xu (2020); Guo et al. (2024) investigated federated DP protocols that can be used when data is kept locally across multiple devices. Adjacent works explore DP versions of PCA in both centralized (Liu et al., 2022) and

federated (Wang and Xu, 2020; Briguglio et al., 2023) settings and claim optimal convergence bounds under distributional assumptions.

Despite these advancements, existing approaches suffer from several limitations. First, their performance heavily depends on the number of singular vectors being computed (Hardt and Price, 2014; Balcan et al., 2016; Guo et al., 2024; Liu et al., 2022; Wang and Xu, 2020), which impacts both utility and privacy guarantees. Second, some are designed for centralized settings (Hardt and Price, 2014; Liu et al., 2022), where a trusted curator is assumed to hold the data. Moreover, some methods (Grammenos et al., 2020; Liu et al., 2022) claim optimality at the cost of strong assumptions about the data distribution (e.g., sub-Gaussianity) which makes these methods harder to use in practice. Some federated versions (Briguglio et al., 2023; Hartebrodt et al., 2024) claim to guarantee privacy due to the federated setting, but it has been shown that decentralization does not offer privacy by design (Geiping et al., 2020). (Dwork et al., 2014; Mangoubi and Vishnoi, 2022) also claims optimality in Differentially Private singular vector estimation but requires materializing, noising, and (eventually) releasing the full covariance matrix and computing its exact SVD decomposition. This is impractical in federated or large-scale settings due to the communication and memory needed and large full-SVD computational cost.

MOD-SuLQ (Chaudhuri et al., 2013) and its federated and streaming PCA variants (Grammenos et al., 2020) offer strong guarantees but are specifically tailored for reconstructing the top principal component ($k = 1$). These methods add noise directly to the covariance matrix, rely on direct, computationally costly exact singular value decompositions (SVD) and are tailored for settings for which the number of samples largely exceeds the dimensionality ($n \gg d$). In contrast, the randomized power method iteratively adds noise directly to the approximation of the singular vectors themselves rather than the input matrix. This strategy reduces the dependence on the dimensionality of the original data, making it computationally more efficient and scalable. Memory-limited, streaming PCA methods such as those proposed by Mitliagkas et al. (2013) are optimized for sequential processing under memory constraints but lack privacy guarantees, requiring additional modifications.

Finally, no fully decentralized versions exist to our knowledge, making them unsuitable for decentralized environments (e.g., recommender systems and social networks), where data is partitioned across users/devices and communications are restricted to a predefined communication graph. The previously introduced versions either use centralized DP Dwork et al. (2014); Hardt and Price (2014); Mangoubi and Vishnoi (2022), which requires a central trusted curator, or local DP (Balcan et al., 2016; Wang and Xu, 2020; Guo et al., 2024), which hinders convergence.

### 1.1 Contributions

We develop Differentially Private (DP) variants of the randomized power method for both centralized and decentralized settings along with their convergence guarantees and empirical benchmarking. Our contributions are:

- **DP randomized power method with tighter sensitivity.** We analyze the $p$-dimensional power method iterates directly and derive an $\ell_2$-sensitivity bound eliminating the explicit $\sqrt{p}$ factor that appears when extrapolating from the one-dimensional case. This leads to a modification in the algorithm allowing for smaller Gaussian noise for the same $(\varepsilon, \delta)$ guarantee. We provide a complete privacy proof via z-CDP composition and its conversion to $(\varepsilon, \delta)$-DP. [1]

- **Convergence guarantees under DP noise.** Using the new sensitivity calibration, we establish a runtime-dependent bound dependent on approximate eigenvector metrics and a runtime-independent bound that replaces this term using matrix coherence quantities. The latter yields a reduced dependence on the iteration rank $p$ compared to previous work.

- **Decentralized variant with distributed DP.** Prior work either assumes a trusted curator for central DP, which is often unrealistic and concentrates control over sensitive data, or relies on local DP, which

---

[1]This proof addresses some mistakes in a privacy proof from Hardt and Roth (2012; 2013) that have been reproduced in several follow-up works (Hardt and Price, 2014; Balcan et al., 2016) and allows for a wider range of DP parameters.

severely degrades utility. Our decentralized design removes the need for a single trusted authority by employing distributed DP with Secure Aggregation (Bell et al., 2020; Bonawitz et al., 2017; Kadhe et al., 2020) or private averaging with correlated noise (Sabater et al., 2022; Allouah et al., 2024). This ensures that individual updates remain private and raw data never leaves local devices. We demonstrate that our method matches the accuracy of centralized DP while providing stronger privacy guarantees, avoiding the high noise levels inherent to local DP and keeping communication overheads low.

- **Empirical evaluation on recommendation data.** We show empirically with a recommender system use-case that the proposed calibration yields considerably lower noise variance and eigenvector approximation error at comparable privacy budgets than prior DP power-method baselines.[2]

The remainder of the paper is organized as follows: In Section 2 we review the necessary background and notation. Section 3 introduces our dataset adjacency notion, derives the improved sensitivity bound (Theorem 3.1), and presents the overall privacy proof (Theorem 3.2) together with runtime-dependent convergence guarantees (Theorem 3.3). In Section 4 we turn this into a fully runtime-independent bound (Theorem 4.1). Section 5 develops the decentralized variant (Algorithm 2), proves its equivalence to the centralized version (Theorem 5.2), and analyzes its communication and computation overhead. We empirically compare the methods on standard recommendation datasets in Section 6.1, and conclude in Section 7 with a discussion of limitations and future directions.

## 2 Background Material and Related Work

**Matrix Norms and Notations.** For any matrix $X \in \mathbb{R}^{n \times m}$, the element-wise maximum norm is defined as $\|X\|_{\max} = \max_{i,j} |X_{ij}|$, where $X_{ij}$ is the $(i,j)$-th element of $X$. The $\ell_2$-norm is denoted as $\|X\|_2$ and the Frobenius norm as $\|X\|_F$. We use $X_{j:}$ to denote the $j$-th row of $X$.

**Eigenvalue Decomposition.** Let $A \in \mathbb{R}^{n \times n}$ be a real-valued symmetric positive semi-definite matrix, where $n$ is a positive integer. The eigenvalue decomposition of $A$ is given by $A = U \Lambda U^\top$, where $U \in \mathbb{R}^{n \times n}$ is a matrix of eigenvectors and $\Lambda \in \mathbb{R}^{n \times n}$ is a diagonal matrix containing corresponding eigenvalues.

**QR Decomposition.** We will use the matrix QR decomposition obtained using the Gram-Schmidt procedure. Given a matrix $X \in \mathbb{R}^{n \times p}$, the QR decomposition factorizes it as $X = QR$, where $Q \in \mathbb{R}^{n \times p}$ is an orthonormal matrix (*i.e.*, $Q^\top Q = I$, where $I$ is the identity matrix) and $R \in \mathbb{R}^{p \times p}$ is an upper triangular matrix.

**Gaussian Random Matrices.** We denote by $\mathcal{N}(\mu, \sigma^2)^{n \times p}$ a $(n \times p)$-dimensional random matrix where each element is an independent and identically distributed random variable, with Gaussian distribution with mean $\mu$ and variance $\sigma^2$.

**Coherence Measures of a Matrix.** We define below two coherence measures for the matrix $A$, which will be useful to state runtime-independent bounds for our method.

- The $\mu_0$-coherence of $A$ is the maximum absolute value of its eigenvectors matrix, defined as $\mu_0(A) = \|U\|_{\max} = \max_{i,j} |U_{ij}|$.

- The $\mu_1$-coherence of $A$ is the maximum row $\ell_2$-norm of its eigenvectors matrix, defined as $\mu_1(A) = \|U\|_{2,\infty} = \max_i (\|U_{i:}\|_2)$.

### 2.1 Differential privacy

With positive integers $n$ and $m$ specifying the matrix dimensions, let $D_1 \in \mathbb{R}^{n \times m}$ and $D_2 \in \mathbb{R}^{n \times m}$ be two matrices representing two datasets embedding sensitive information. $D_1$ and $D_2$ are said to be adjacent ($D_1 \sim D_2$) if they differ on one sensitive element of the dataset (granularity can differ by application). For

---

[2]We provide an anonymous code repository here.

example, in a recommender system application, $D_1$ and $D_2$ can be binary user-item interaction matrices and a sensitive element of the dataset can be a user-item interaction corresponding to an entry in the matrices.

A randomized algorithm $\mathcal{M}$ is $(\epsilon, \delta)$-Differential Private (DP) if for all adjacent datasets $D_1$ and $D_2$, and for all measurable subsets $S \subseteq \text{Range}(\mathcal{M})$, the following holds:

$$\Pr[\mathcal{M}(D_1) \in S] \le e^\epsilon \Pr[\mathcal{M}(D_2) \in S] + \delta, \tag{1}$$

where $\epsilon$ is a small positive scalar representing the maximum privacy loss (smaller values indicate stronger privacy guarantees), and $\delta$ is a (typically chosen to be negligible) probability that the privacy guarantee fails.

Let $f : \mathbb{R}^{n \times m} \to \mathbb{R}^d$ be a query function associated with a mechanism $\mathcal{M}$. DP guarantees of mechanisms are defined using the sensitivity of $f$. In our contribution we use the $\ell_2$-sensitivity, denoted by $\Delta_2(f)$ or $\Delta_2$ and defined as $\Delta_2 = \max_{D_1 \sim D_2} \|f(D_1) - f(D_2)\|_2$.

## 2.2 Privacy-Preserving Randomized Power Method

Let $k \in \mathbb{N}^*$ be the target rank, let $b \in \mathbb{N}^*$ be a small positive integer, and let $\eta \in \mathbb{R}^*$ be an approximation tolerance. Let $\boldsymbol{A} \in \mathbb{R}^{n \times n}$ be a symmetric positive semi-definite (PSD) matrix, $p = k + b$ and $L$ be the number of power iterations. The aim of the randomized power method is to construct a matrix $\boldsymbol{X}^L \in \mathbb{R}^{n \times p}$ whose column space approximates that of the top-$k$ eigenvectors of $\boldsymbol{A}$, $i.e.$, $\boldsymbol{U}_k \in \mathbb{R}^{n \times k}$. Specifically, it aims to satisfy

$$\left\| \boldsymbol{U}_k - \boldsymbol{X}^L (\boldsymbol{X}^L)^\top \boldsymbol{U}_k \right\| \le \eta. \tag{2}$$

To protect sensitive user information embedded in $\boldsymbol{A}$ while computing $\boldsymbol{X}^L$, prior work (Hardt and Price, 2014; Balcan et al., 2016) showed that the randomized power method can be implemented as Algorithm 1 to be $(\epsilon, \delta)$-Differential Private, with adjacency defined as a single element change in $\boldsymbol{A}$. The corresponding adjacency notion is defined in more detail in Equation 6.

---

**Algorithm 1** Privacy-preserving randomized power method

1: **Input**: Matrix $\boldsymbol{A} \in \mathbb{R}^{n \times n}$, number of iterations $L$, target rank $k$, iteration rank $p \ge k$, privacy parameters $\epsilon, \delta$
2: **Output**: approximated eigen-space $\boldsymbol{X}^L \in \mathbb{R}^{n \times p}$, with orthonormal columns.
3: **Initialization**: orthonormal $\boldsymbol{X}^0 \in \mathbb{R}^{n \times p}$ by QR decomposition on a random Gaussian matrix $\boldsymbol{G}_0$; noise variance parameter $\sigma = \epsilon^{-1} \sqrt{4L \log(1/\delta)}$;
4:
5: **for** $\ell = 1$ to $L$ **do**
6:     Compute $\boldsymbol{Y}_\ell = \boldsymbol{A} \boldsymbol{X}^{\ell-1} + \boldsymbol{G}_\ell$ with $\boldsymbol{G}_\ell \sim \mathcal{N}(0, \sigma_l^2 = \Delta_l^2 \cdot \sigma^2)^{n \times p}$;
7:     Compute QR factorization $\boldsymbol{Y}_\ell = \boldsymbol{X}^\ell \boldsymbol{R}_\ell$;
8: **end for**

---

To compute the standard deviation of the noise required to satisfy $(\epsilon, \delta)$-DP, one needs to bound the $\ell_2$-sensitivity $\Delta_l$ of the revealed outputs at each iteration $l$ of the algorithm. Here, $\Delta_l$ bounds the change on $\boldsymbol{A} \boldsymbol{X}^{l-1}$ under a single-element perturbation in the sensitive data matrix $\boldsymbol{A}$. Prior works (Hardt and Price, 2014; Balcan et al., 2016) use the estimate

$$\Delta_l^{\text{prior}} \triangleq \sqrt{p} \|\boldsymbol{X}^l\|_{\max}, \tag{3}$$

which upper-bounds the true sensitivity, i.e., $\Delta_l \le \Delta_l^{\text{prior}}$.

Using this bound[3], the standard deviation $\sigma_l$ of the Gaussian noise added at iteration $l$ to achieve $(\epsilon, \delta)$-DP is $\sigma_l = \sqrt{p} \left\| \boldsymbol{X}^l \right\|_{\max} \epsilon^{-1} \sqrt{4L \ln(1/\delta)}$, where $L$ is the total number of power iterations.

---

[3]See Fig. 3 of Hardt and Price (2014), Alg. 2 of Balcan et al. (2016), and Theorem 6 of Guo et al. (2024).

### 2.3 Existing convergence bounds

To our knowledge, the strongest convergence bound for the privacy-preserving randomized power method (Algorithm 1) uses the bound $\Delta_l \leq \Delta_l^{prior}$ and is given by Balcan et al. (2016) in their Corollary 3.1. The following theorems rely on the conditions in Assumption 2.1, which are enforced by judicious choice of the added noise.

**Assumption 2.1.** Let $\boldsymbol{A} \in \mathbb{R}^{n \times n}$ be a symmetric matrix. Fix a target rank $k$, an intermediate rank $q \geq k$, and an iteration rank $p$, with $k \leq q \leq p$. Let $\boldsymbol{U}_q \in \mathbb{R}^{n \times q}$ be the top-$q$ eigenvectors of $\boldsymbol{A}$ and let $\lambda_1 \geq \cdots \geq \lambda_n \geq 0$ denote its eigenvalues. Let us fix $\eta = O\left(\frac{\lambda_q}{\lambda_k} \cdot \min\left\{\frac{1}{\log\left(\frac{\lambda_k}{\lambda_q}\right)}, \frac{1}{\log(\tau n)}\right\}\right)$.

Assume that at every iteration $l$ of Algorithm 1, $\boldsymbol{G}_\ell$ satisfies, for some constant $\tau > 0$:

$$\|\boldsymbol{G}_\ell\|_2 = O\left(\eta(\lambda_k - \lambda_{q+1})\right), \quad \text{and} \quad \|\boldsymbol{U}_q^\top \boldsymbol{G}_\ell\|_2 = O\left(\eta\left(\lambda_k - \lambda_{q+1}\right) \frac{\sqrt{p} - \sqrt{q-1}}{\tau \sqrt{n}}\right). \tag{4}$$

We now restate the Private Power Method major result in (Balcan et al., 2016).

**Theorem 2.2** (Private Power Method (PPM), reduction to $s = 1$ from the proof in Appendix C.1 from Balcan et al. (2016)). *Let $\boldsymbol{A} \in \mathbb{R}^{n \times n}$ be a symmetric data matrix. Fix target rank $k$, intermediate rank $q \geq k$, and iteration rank $p$, with $2q \leq p \leq n$. Suppose the number of iterations $L$ is set as $L = \Theta(\frac{\lambda_k}{\lambda_k - \lambda_{q+1}} \log(n))$. Let $\epsilon, \delta \in (0, 1)$ be the differential privacy parameters. Let $\boldsymbol{U}_q \in \mathbb{R}^{n \times q}$ be the top-$q$ eigenvectors of $\boldsymbol{A}$ and let $\lambda_1 \geq \cdots \geq \lambda_n \geq 0$ denote its eigenvalues. Then Algorithm 1 with $\Delta_l = \Delta_l^{prior}$ is ($\epsilon$, $\delta$)-DP and with probability at least 0.9*

$$\|(\boldsymbol{I} - \boldsymbol{X}^L(\boldsymbol{X}^L)^\top)\boldsymbol{U}_k\|_2 \leq \eta \quad and \quad \|(\boldsymbol{I} - \boldsymbol{X}^L(\boldsymbol{X}^L)^\top)\boldsymbol{A}\|_2^2 \leq \lambda_{k+1}^2 + \eta^2 \lambda_k^2 \tag{5}$$

$$with \quad \eta = O\left(\frac{\epsilon^{-1} \max_l\left(\|\boldsymbol{X}^l\|_{\max}\right)\sqrt{4pLn\log(1/\delta)\log(L)}}{\lambda_k - \lambda_{q+1}}\right)$$

$$and\ also \quad \eta = O\left(\frac{\epsilon^{-1} \|\boldsymbol{U}\|_{\max}\sqrt{4pLn\log(1/\delta)\log(n)\log(L)}}{\lambda_k - \lambda_{q+1}}\right).$$

## 3 Proposed Differentially Private Power Method Convergence Bounds

In this section, we present a noise-reduced Differentially Private Power Method. First, we introduce a generalized definition of adjacency that goes beyond single-entry changes in a PSD matrix to allow directly for more applications. Then, we modify the Private Power Method to calibrate DP noise using a new, tighter sensitivity bound, eliminating the $\sqrt{p}$ (associated to target rank) factor in prior work. We then show analytically that this refinement yields sharper convergence guarantees.

**Adjacency notion.** In prior work, Hardt and Price (2014) consider symmetric matrices, while Balcan et al. (2016) restrict to positive semi-definite (PSD) matrices. In practice, the matrices of interest are often covariance matrices (Mangoubi and Vishnoi, 2022; Mitliagkas et al., 2013; Hardt and Price, 2014)[4] , which are symmetric and PSD. Concretely, two datasets represented by symmetric (resp. PSD) matrices $\boldsymbol{A}, \boldsymbol{A}' \in \mathbb{R}^{n \times n}$ are considered adjacent (denoted $\boldsymbol{A} \sim \boldsymbol{A}'$) if they differ in a single entry with a Frobenius norm difference of at most 1. We can therefore write

$$\boldsymbol{A}' = \boldsymbol{A} + c \cdot \boldsymbol{e}_i \boldsymbol{e}_j^\top, \tag{6}$$

where $\boldsymbol{e}_i, \boldsymbol{e}_j \in \mathbb{R}^n$ are canonical basis vectors, $c \leq 1$ represents the magnitude of the change, and $0 \leq i, j < n$. This adjacency notion models a sensitive change as a modification to a single element in $\boldsymbol{A}$ to protect individual

---

[4]More generally, for a non-symmetric data matrix $\boldsymbol{A} \in \mathbb{R}^{n \times m}$, the covariance matrix $\boldsymbol{B} = \boldsymbol{A}\boldsymbol{A}^\top$ is symmetric PSD and has the same left singular vectors as $\boldsymbol{A}$. Hence, approximating the top eigenvectors of $\boldsymbol{B}$ is equivalent to approximating the top left singular vectors of $\boldsymbol{A}$. Furthermore, Halko et al. (2011) show that applying the randomized power method to $\boldsymbol{B}$ improves the convergence rate from a multiplicative eigenvalue gap factor $(\sigma_k/\sigma_{k+1})^q$ to $(\sigma_k/\sigma_{k+1})^{2q}$, accelerating convergence.

element-wise updates under Differential Privacy. However, since $\boldsymbol{A}$ is symmetric positive semi-definite and changes must preserve this property, this formulation restricts updates to be diagonal, hindering possible applications.

We propose a new, more general, notion of adjacency to allow for other types of updates:

$$\boldsymbol{A}' = \boldsymbol{A} + \boldsymbol{C}, \tag{7}$$

where $\boldsymbol{C} \in \mathbb{R}^{n \times n}$ is a symmetric matrix representing the update, subject to $\sqrt{\sum_{i=1}^{n} \|\boldsymbol{C}_{i,:}\|_1^2} \leq 1$.

The proposed adjacency notion is strictly more general than Equation 6. For example, setting $\boldsymbol{C} = c \cdot \boldsymbol{e}_i \boldsymbol{e}_j^\top$ recovers the original definition. $\boldsymbol{C}$ can have non-zero entries on the diagonal, anti-diagonal, or any symmetric pattern, allowing directly for a variety of updates maintaining symmetry.

In the context of recommender systems, where $\boldsymbol{A} = \boldsymbol{R}^\top \boldsymbol{R}$ represents the item-item similarity matrix and $\boldsymbol{R} \in \mathbb{R}^{m \times n}$ is the user-item interaction matrix, our proposed notion of adjacency enables element-wise modifications in $\boldsymbol{R}$ (*i.e.*, protecting individual user-item interactions). Such changes in $\boldsymbol{R}$ propagate to multiple elements of $\boldsymbol{A}$, which could not be adequately[5] accounted for under the previous adjacency definition (Equation 6). By adopting our more general adjacency definition, we make our privacy guarantees applicable to a wider range of real-world scenarios.

**Sensitivity bound.** The previous sensitivity bound $\Delta_l^{\text{prior}}$ (Hardt and Price, 2014; Balcan et al., 2016) for $\boldsymbol{A}\boldsymbol{X}^{l-1}$ defined in Equation 3 was estimated by extrapolating from the case where $\boldsymbol{X}^{l-1} \in \mathbb{R}^{n \times 1}$ to the general case $\boldsymbol{X}^{l-1} \in \mathbb{R}^{n \times p}$, leading to a dependence on $\sqrt{p}$. This leads to an overestimation of the sensitivity and to unnecessarily large noise addition. We directly analyze the change in $\boldsymbol{X}^{l-1} \in \mathbb{R}^{n \times p}$ and derive a tighter bound on the sensitivity $\Delta_l$. By using our adjacency notion from Equation 7, we establish the following result (proof in Appendix A.1):

**Theorem 3.1** (Improved Sensitivity Bound)**.** *Let $\boldsymbol{A}'$ be defined as in Equation 7, and consider the sensitivity $\Delta_l = \sup_{\boldsymbol{A} \sim \boldsymbol{A}'} \|\boldsymbol{A}'\boldsymbol{X}^l - \boldsymbol{A}\boldsymbol{X}^l\|_F$. Then,*

$$\Delta_l \leq \max_i \|\boldsymbol{X}_{i:}^l\|_2 \triangleq \hat{\Delta}_l. \tag{8}$$

**Note.** Since $\|\boldsymbol{X}_{i:}^l\|_2 \leq \sqrt{p}\|\boldsymbol{X}^l\|_{\max}$, our sensitivity bound is always tighter or equal to the prior bound.

**Privacy proof.** Our algorithm ensures $(\epsilon,\delta)$-Differential Privacy by adding calibrated Gaussian noise at each iteration of the power method. The overall privacy guarantee across iterations is then derived using results from adaptive composition of DP mechanisms, as initially proposed in (Bun and Steinke, 2016). To clarify the ambiguities or errors present in the previous privacy proofs (see Appendix A.2), and ensure that our privacy guarantees are met, we propose a result with a new proof of the Differential Privacy guarantees for our overall algorithm in Theorem 3.2, whose derivation is in Appendix A.2.3.

**Theorem 3.2** (Privacy proof for the PPM)**.** *Let $\delta \in (0,1)$ and $\epsilon > 0$ such that $\delta \leq \exp\left(-\frac{\epsilon}{4}\right)$. Then, Algorithm 1 with $\Delta_l = \max_i \|\boldsymbol{X}_{i:}^l\|_2$ is $(\epsilon, \delta)$-Differentially Private.*

**Improved convergence bound.** Building on Equation 8, we present in Theorem 3.3 a tighter convergence bound than the one proposed in (Balcan et al., 2016). Additionally, unlike past proofs (Hardt and Price, 2014; Balcan et al., 2016), our proposed privacy proof (given in Theorem 3.2) does not restrict $\epsilon \leq 1$. We provide the proof in Appendix A.3.

**Theorem 3.3** (Improved PPM with Runtime-Dependent Bound)**.** *Let $\boldsymbol{A} \in \mathbb{R}^{n \times n}$ be a symmetric data matrix. Fix target rank $k$, intermediate rank $q \geq k$ and iteration rank $p$ with $2q \leq p \leq n$. Suppose the number of iterations $L = \Theta(\frac{\lambda_k}{\lambda_k - \lambda_{q+1}} \log(n))$. Let $\delta \in (0,1)$ and $\epsilon > 0$ be privacy parameters such that $\delta \leq \exp\left(-\frac{\epsilon}{4}\right)$. Let $\boldsymbol{U}_k \in \mathbb{R}^{n \times k}$ be the top-$k$ eigenvectors of $\boldsymbol{A}$ and let $\lambda_1 \geq \cdots \geq \lambda_n \geq 0$ denote its eigenvalues. Then Algorithm 1 is $(\epsilon,\delta)$-DP with $\Delta_l = \max_i \|\boldsymbol{X}_{i:}^l\|_2$ and with probability at least 0.9*

$$\|(\boldsymbol{I} - \boldsymbol{X}^L(\boldsymbol{X}^L)^\top)\boldsymbol{U}_k\|_2 \leq \eta \quad and \quad \|(\boldsymbol{I} - \boldsymbol{X}^L(\boldsymbol{X}^L)^\top)\boldsymbol{A}\|_2^2 \leq \lambda_{k+1}^2 + \eta^2 \lambda_k^2 \tag{9}$$

---

[5]Such accounting would require additional steps using for instance group privacy.

$$with \quad \eta = O\left(\frac{\epsilon^{-1}\max_{i,l}\|\boldsymbol{X}_{i:}^l\|_2\sqrt{Ln\log(1/\delta)\log(L)}}{\lambda_k - \lambda_{q+1}}\right). \tag{10}$$

## 4 Proposed Runtime-Independent Convergence Bound

We presented in Theorem 3.3 a convergence bound involving $\max_i \|\boldsymbol{X}_{i:}\|_2$, which is only observable during the execution of the algorithm. To provide a more practical analysis, we now derive a runtime-independent convergence bound in Theorem 4.1 by careful bounding of $\max_i \|\boldsymbol{X}_{i:}\|_2$. We provide a proof in Appendix A.4. This bound makes it possible to have a tight analysis in two regimes:

- **Small $\mu_0$-coherence, small $p$:** Previously proposed in Balcan et al. (2016), this bound is useful in a regime with small $\mu_0$ when computing few eigenvectors, with a dependence on $\sqrt{p\log(n)} \cdot \mu_0(\boldsymbol{A})$.

- **Larger $\mu_0$ or $p$:** We propose a new bound tailored for the multi-dimensional power method, depending on $\mu_1(\boldsymbol{A})$, with a reduced dependence on the number of eigenvectors $p$.

We note that we are likely to be in the second regime in practice, as we highlight in Section 6.1.

**Theorem 4.1.** *Improved PPM with Runtime-Independent Bound.* *Let $\boldsymbol{A} \in \mathbb{R}^{n\times n}$ be a symmetric data matrix. Fix target rank $k$, intermediate rank $q \geq k$ and iteration rank $p$ with $2q \leq p \leq n$. Suppose the number of iterations $L$ is set as $L = \Theta(\frac{\lambda_k}{\lambda_k - \lambda_{q+1}}\log(n))$. Let $\boldsymbol{U}_q \in \mathbb{R}^{n\times q}$ be the top-$q$ eigenvectors of $\boldsymbol{A}$ and let $\lambda_1 \geq \cdots \geq \lambda_n \geq 0$ denote its eigenvalues. Let $\delta \in (0,1)$ and $\epsilon > 0$ be privacy parameters such that $\delta \leq \exp\left(-\frac{\epsilon}{4}\right)$. Then Algorithm 1 is $(\epsilon,\delta)$-DP and we have with probability at least 0.9*

$$\|(\boldsymbol{I} - \boldsymbol{X}^L(\boldsymbol{X}^L)^\top)\boldsymbol{U}_k\|_2 \leq \eta \quad and \quad \|(\boldsymbol{I} - \boldsymbol{X}^L(\boldsymbol{X}^L)^\top)\boldsymbol{A}\|_2^2 \leq \lambda_{k+1}^2 + \eta^2\lambda_k^2 \tag{11}$$

$$with \quad \eta = O\left(\frac{\epsilon^{-1}\cdot\min(\mu_0(\boldsymbol{A})\sqrt{p\cdot\log(n)},\mu_1(\boldsymbol{A}))\cdot\sqrt{Ln\log(1/\delta)\log(L)}}{\lambda_k - \lambda_{q+1}}\right). \tag{12}$$

## 5 Decentralized version

A natural approach to Differentially Private data analysis is to centralize the data on a trusted server and apply a centralized DP mechanism. However, this requires full trust in the curator, which is not always acceptable in practice. Relying on a single trusted party raises both technical and governance concerns: it concentrates sensitive information in one location and further reinforces the dependence on large data custodians. By contrast, our decentralized approach, based on Secure Aggregation or private averaging, avoids exposing raw data to a central server and distributes trust across participants. We demonstrate that this decentralization comes at no loss in accuracy compared to centralized DP, while providing strengthened privacy guarantees under the same threat model.

In this section, we consider a decentralized setting in which the matrix $\boldsymbol{A}$ is distributed across multiple clients. Specifically, each client $i$ holds a private matrix $\boldsymbol{A}^{(i)} \in \mathbb{R}^{n\times n}$, such that the global matrix is the sum of these local matrices: $\boldsymbol{A} = \sum_{i=1}^s \boldsymbol{A}^{(i)}$. The goal is for the clients to collaboratively compute an orthonormal basis for the range of $\boldsymbol{A}$, similar to the centralized randomized power method, but without revealing their individual private matrices $\boldsymbol{A}^{(i)}$ to the server or to other clients.

The randomized power method involves linear operations, making it well-suited for parallelization and distributed computation. Balcan et al. (2016); Guo et al. (2024) proposed private and federated power methods using communication over public channels between clients and a server. However, these approaches rely on Local Differential Privacy, since the data exchanged can be observed by everyone, which requires high levels of noise to ensure privacy.

To enhance privacy while retaining the benefits of distributed computation, we propose to integrate Secure Aggregation into the method, a lightweight Multi-Party Computation protocol. Secure Aggregation allows clients to collaboratively compute sums without revealing individual data, enabling the use of distributed

DP. This approach offers privacy guarantees similar to central DP and eliminates the need for a trusted curator. Distributed DP has been extensively studied in the literature (Goryczka et al., 2013; Ghazi et al., 2019; Kairouz et al., 2021; Chen et al., 2021; Wei et al., 2024).

In distributed DP, each client adds carefully calibrated noise to their local contributions before participating in the Secure Aggregation protocol. The variance of the noise is chosen such that the sum of locally added noises across clients has a variance comparable to that used in central DP, thereby achieving similar Differential Privacy guarantees without requiring a trusted aggregator, and allowing clients to keep data locally.

We operate under the **honest-but-curious** threat model, where clients follow the protocol correctly but may attempt to learn information from received data. We also assume that there are no dropout users during the computation, and that the result of Secure Aggregation is revealed to every party at each iteration. For simplicity, this paper neglects the effects of data quantization or errors introduced by modular arithmetic in distributed DP and refer to Kairouz et al. (2021) for a more technical implementation taking this into account.

It is also possible to use a fully decentralized DP protocol to perform secure averaging without relying on a central server, as demonstrated by Sabater et al. (2022). Adopting such approaches can further enhance decentralization while maintaining similar utility and communication costs for the method.

### 5.1 Decentralized and private power method using distributed DP:

We introduce a federated version of the Privacy-Preserving Power Method in Algorithm 2. This version significantly reduces the noise variance by a factor of $sp \log n \|\boldsymbol{U}\|_\infty^2$ compared to the method in Balcan et al. (2016). The improvement comes from the fact that we emulate centralized Differential Privacy (DP) over secure channels, avoiding the higher noise required in local DP settings with public channels, and that we use our generally tighter sensitivity bound.

We now give a simplified definition of Secure Aggregation and demonstrate the equivalence between Algorithm 2 and our centralized Algorithm 1 in Theorem 5.2 (proof in Appendix A.4).

**Definition 5.1.** Let $SecAgg(\boldsymbol{Y}_\ell^{(i)}, \{i|1 \le i \le s\})$ the Secure Aggregation of matrices $\boldsymbol{Y}_\ell^{(i)}$ held by users indexed by $\{i|1 \le i \le s\}$. It is equivalent to computing $\boldsymbol{Y}_\ell = \sum_{i=1}^s \boldsymbol{Y}_\ell^{(i)}$ over secure channels.

---

**Algorithm 2** Federated private power method

1: **Input**: distributed matrices $\boldsymbol{A}^{(1)}, \cdots, \boldsymbol{A}^{(s)} \in \mathbb{R}^{n \times n}$, number of iterations $L$, target rank $k$, iteration rank $p \ge k$, privacy parameters $\epsilon, \delta$.
2: **Output**: approximated eigen-space $\boldsymbol{X}^L \in \mathbb{R}^{n \times p}$, with orthonormal columns.
3: **Initialization**: orthonormal $\boldsymbol{X}^0 \in \mathbb{R}^{n \times p}$ by QR decomposition on a random Gaussian matrix $\boldsymbol{G}_0$; noise variance parameter $\nu = \epsilon^{-1}\sqrt{\frac{4L \log(1/\delta)}{s}}$;
4:
5: **for** $\ell = 1$ to $L$ **do**
6:     The central node broadcasts $\boldsymbol{X}^{\ell-1}$ to all $s$ computing nodes;
7:     Computing node $i$ computes $\boldsymbol{Y}_\ell^{(i)} = \boldsymbol{A}^{(i)}\boldsymbol{X}^{\ell-1} + \boldsymbol{G}_\ell^{(i)}$ with $\boldsymbol{G}_\ell^{(i)} \sim \mathcal{N}(0, \Delta_l^2\nu^2)^{n \times p}$;
8:     The central node computes with the clients $\boldsymbol{Y}_\ell = SecAgg(\boldsymbol{Y}_\ell^{(i)}, \{i|1 \le i \le s\})$;
9:     The central node computes QR factorization $\boldsymbol{Y}_\ell = \boldsymbol{X}^\ell \boldsymbol{R}_\ell$;
10: **end for**

---

**Theorem 5.2** (Privacy and utility of Algorithm 2). *The Decentralized Privacy-Preserving Power Method (Algorithm 2) provides the same privacy guarantees and achieves equivalent utility (in terms of convergence) as its centralized version (Algorithm 1).*

### 5.2 Cost analysis:

A central distinction between the different methods lies in whether they require explicit materialization of the covariance matrix. Classical approaches such as AnalyzeGauss (Dwork et al., 2014) and its re-

analysis (Mangoubi and Vishnoi, 2022) construct, noise, and release the full covariance matrix and assume that the user can compute its exact SVD. While this yields strong theoretical guarantees, it is impractical in large-scale or federated settings where the input matrix is distributed, as both the memory and communication overhead scale quadratically in the number of features due to the covariance matrix computation and communication.

In contrast, methods based on the randomized power method such as DistPrivPCA (Balcan et al., 2016), FedPower (Guo et al., 2024), and our proposed decentralized PPM never require forming the full covariance matrix. Instead, they operate through repeated sparse matrix-vector products combined with dense Gaussian noise addition applied only to the evolving eigenvector estimates of dimensions $n \times p$. Since $p \ll n$ in practice, this strategy dramatically reduces both the communication and memory burden while still ensuring differential privacy.

Table 1 summarizes the overall communication costs. DistPrivPCA and FedPower both involve transmitting $np$-dimensional updates from each client to the server at each round, leading to $O(Lnp)$ total communication per client and $O(Lsnp)$ at the server. AnalyzeGauss incurs a prohibitive $O(n^2)$ transmission cost per client. We note that our decentralized variant integrates Secure Aggregation into the randomized power method. This introduces an additional logarithmic communication factor $\log(s)$ per round, but since $\log(s) \ll np$ in practical regimes (e.g., for MovieLens-10M, $n = 10,677$, $p = 32$, $L = 2$, while $\log(s) \approx 5$), the asymptotic communication costs are dominated by the same $np$ terms as in prior work.

Overall, our approach matches the low-dimensional communication footprint of FedPower and DistPrivPCA, while providing stronger privacy through distributed DP and avoiding the impractical explicit covariance materialization of AnalyzeGauss, which requires $n^2$ memory.

Table 1: Communication complexities comparison across methods.

|  | Client | Server |
|---|---|---|
| DistPrivPCA (Balcan et al., 2016) FedPower (Guo et al., 2024) | $Lnp$ | $Ls\,np$ |
| AnalyzeGauss (Dwork et al., 2014) | $n^2$ | $sn^2$ |
| Proposed | $L\,[np + \log(s)]$ | $Ls\,[np + \log(s)]$ |

## 6 Empirical comparison of the proposed bounds

We introduced runtime-dependent and runtime-independent convergence bounds for our algorithm. The runtime-dependent bound depends on $\boldsymbol{A}$ and the iteratively computed $\boldsymbol{X}^l$, whereas the runtime-independent bound (Theorem 4.1) only depends on $\boldsymbol{A}$. To illustrate the practical impact of our changes, we focus on an application in recommender systems and compare how our algorithms perform in this context. Additionally, we use a statistical approximation to see how the bounds compare at the first step of the algorithm (sketching step), regardless of the application in Section 6.2.

### 6.1 Application to recommender systems

State-of-the-art recommender systems like GF-CF (Shen et al., 2021) and BSPM (Choi et al., 2023) utilize singular or eigenvalue decomposition as part of their algorithms. Specifically, they represent the dataset (user-item interactions) using a normalized adjacency matrix $\tilde{\boldsymbol{R}}$, from which they compute a normalized item-item matrix $\tilde{\boldsymbol{P}}$. To get rid of noise in $\tilde{\boldsymbol{R}}$, they compute an ideal low-pass filter based on the top-$p$ eigenvectors of $\tilde{\boldsymbol{P}}$ and apply it to $\tilde{\boldsymbol{R}}$. Detailed definitions are provided in Appendix B.

The first four columns of Table 2 present key statistics of popular recommendation datasets used in the literature. Depending on the use case, these datasets may be held by a central curator, or distributed among the users of a recommender system. Accordingly, either the centralized PPM (Algorithm 1) or the decentralized PPM (Algorithm 2) can be used to compute the leading eigenvectors. To demonstrate the practicality of our proposed methods, we focus on a decentralized setting where each user has access to its own

ratings and collaborates with other users to compute the desired eigenvectors. $\Delta_l^{prior} = \mu_0(\boldsymbol{A})\sqrt{p \cdot \log(n)}$ denotes the sensitivity bound proposed in (Balcan et al., 2016) while $\hat{\Delta}_l = \mu_1(\boldsymbol{A})$ uses our multidimensional refinement.

Table 2: Statistics of datasets

| Dataset | Users | Items | Interactions | $\mu_0$ | $\mu_1$ | $\Delta_l^{prior}$ (prior) | $\hat{\Delta}_l$ (ours) |
|---|---|---|---|---|---|---|---|
| Amazon-book | 52,643 | 91,599 | 2,984,108 | 0.33 | 0.99 | $1.12 \times \sqrt{p}$ | 0.99 |
| MovieLens | 71,567 | 10,677 | 7,972,582 | 0.39 | 1.00 | $1.19 \times \sqrt{p}$ | 1.00 |
| EachMovie | 74,425 | 1,649 | 2,216,887 | 0.49 | 1.00 | $1.33 \times \sqrt{p}$ | 1.00 |
| Jester | 54,906 | 151 | 1,450,010 | 0.73 | 1.00 | $1.64 \times \sqrt{p}$ | 1.00 |

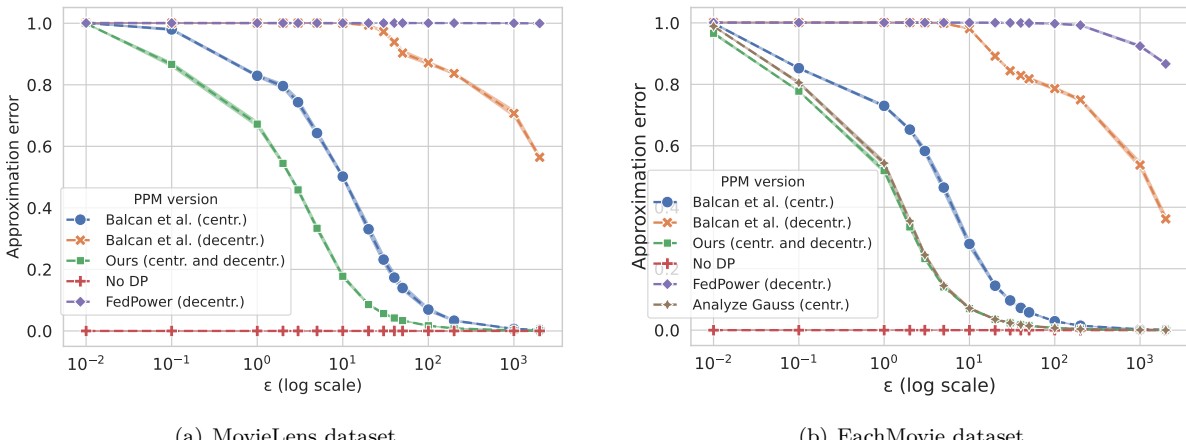

(a) MovieLens dataset.  (b) EachMovie dataset.

Figure 1: Comparison of the impact of the Differential Privacy parameter $\epsilon$ on the relative approximation error $\frac{\|\tilde{\boldsymbol{R}}_p - \boldsymbol{R}_p\|_F}{\|\boldsymbol{R}_p\|_F}$, where $\tilde{\boldsymbol{R}}_p$ represents the approximated matrix and $\boldsymbol{R}_p$ the original matrix. Results are means on 10 runs with shaded bands indicating 99% confidence intervals (computed via bootstrap), shown for the MovieLens and EachMovie datasets, with $p = 32$, $L = 3$ and $\delta = 10^{-4}$.

**Empirical comparison of the runtime-independent bounds.**   As discussed in Section 4, our proposed runtime-independent bound is tighter than those in Hardt and Price (2014); Balcan et al. (2016) when $\mu_1(\boldsymbol{A}) \leq \mu_0(\boldsymbol{A})\sqrt{p \cdot \log(n)}$, where $\boldsymbol{U}$ denotes the eigenvectors of $\boldsymbol{A}$. The last two columns from Table 2 show that this condition holds for popular recommender system datasets. As we can deduce from Table 2, the proposed bound theoretically allows us to converge to solutions with much smaller $\eta$ values, especially when the desired number of factors $p$ is large.

**Practical utility of the proposed algorithms.**   We saw that the proposed runtime-independent bounds were practically tighter than the previous one for the task of interest. We now demonstrate that Algorithm 2 can be used to compute the top-$p$ eigenvectors $\boldsymbol{U}_p$ of the item-item matrix $\tilde{\boldsymbol{P}}$ under $(\epsilon, \delta)$-DP (proof in Appendix B):

**Lemma 6.1.** *Algorithm 2 with $\Delta_l = \sqrt{2}\max_i \|\boldsymbol{X}_{i:}^l\|_2$ can approximate $\boldsymbol{U}_p$, the top-p eigenvectors of $\tilde{\boldsymbol{P}}$ in a decentralized setting under $(\epsilon, \delta)$-Differential Privacy.*

Both GF-CF (Shen et al., 2021) and BSPM (Choi et al., 2023) use $\boldsymbol{U}_p$ to compute the ideal low-pass filter and apply it to the interaction matrix $\boldsymbol{R}$, yielding $\boldsymbol{R}_p$. To illustrate the practicality of our proposed method, we compute approximations of $\boldsymbol{U}_p$ (denoted by $\tilde{\boldsymbol{U}}_p$) using either our decentralized PPM, the previous PPM versions in  Balcan et al. (2016), FedPower from Guo et al. (2024) or the centralized Analyze Gauss method (Dwork et al., 2014; Mangoubi and Vishnoi, 2022). We then use $\tilde{\boldsymbol{U}}_p$ to compute an approximation of

$\boldsymbol{R}_p$ (denoted by $\tilde{\boldsymbol{R}}_p$), and compute the relative approximation error $\frac{\|\tilde{\boldsymbol{R}}_p - \boldsymbol{R}_p\|_F}{\|\boldsymbol{R}_p\|_F}$ associated to each version. We provide experimental details in Appendix B.2.

Figure 1 illustrates the impact of the privacy parameter $\epsilon$[6] on the approximation error for the EachMovie and MovieLens datasets, with $p = 32$, $L = 3$ and $\delta = 10^{-4}$, clearly showing the advantage of our method over prior decentralized methods. Our decentralized PPM achieves relative approximation error of $\approx 1/10$ for values of $\epsilon \in (5, 10)$ for EachMovie and $\epsilon \approx 20$ for MovieLens. In contrast, other decentralized methods (Balcan et al., 2016; Guo et al., 2024) require $\epsilon$ to be of the order of at least $10^3$ to achieve comparable errors, which does not seem to provide meaningful privacy protection[7]. (Guo et al., 2024) yields the worst relative approximation errors. We hypothesize that this is because it uses worst case, non-adaptive sensitivity bounds for DP, as opposed to our proposed method and those of Balcan et al. (2016). For a fixed approximation error, both our centralized and decentralized methods yield $\epsilon$ roughly four times smaller than required by the centralized method of Balcan et al. (2016), demonstrating that our propositions significantly strengthen privacy guarantees for a target approximation error. We also see that our decentralized method performs on par with the centralized Analyze Gauss (Dwork et al., 2014; Mangoubi and Vishnoi, 2022) on EachMovie and Jester, and we recall that Analyze Gauss would incur prohibitive communication and memory overhead in a decentralized setting (See Section 5.2). Due to its high memory requirements, we did not benchmark Analyze Gauss on MovieLens-10m. We additionally evaluate the impact of the number of eigenvectors on the performance of the different methods in Appendix B.3.

## 6.2 Matrix-agnostic comparison of the runtime-dependent bounds

We analyze the tightness of our proposed runtime-dependent bound compared to the previous bound in a matrix-agnostic manner, focusing on the first iteration of the algorithm.

At iteration $l = 0$, the leading singular vectors are initialized as $\boldsymbol{X}^0 = \mathrm{Q}(\boldsymbol{\Omega})$, where $\boldsymbol{\Omega} \sim \mathcal{N}(0, 1)^{n \times k}$ and $\mathrm{Q}(\boldsymbol{\Omega})$ denotes the orthonormal matrix obtained from the QR decomposition of $\boldsymbol{\Omega}$. Since $\boldsymbol{X}^0$ is independent of the matrix $\boldsymbol{A}$, it allows us to assess the relative tightness of the proposed bound $\hat{\Delta}_0$ compared to the previous bound $\Delta_0^{prior}$ at the first step. We note that when the randomized power method runs for only one step, it is akin to sketching (Halko et al., 2011).

Let $r(k, n)$ be the ratio of the expected values of the two bounds, as a function of the desired number of eigenvectors $k$ and the matrix dimension $n$, i.e. $r(k, n) = \frac{\mathbb{E}[\Delta_0^{prior}]}{\mathbb{E}[\hat{\Delta}_0]}$.

**Theoretical approximation of $r(k, n)$.** We derive an asymptotic approximation of $r(k, n)$ using Approximation 6.2, whose derivation is in Appendix C.

**Approximation 6.2.** Let $\boldsymbol{X}^0 = \mathrm{Q}(\boldsymbol{\Omega})$ where $\boldsymbol{\Omega} \sim \mathcal{N}(0, 1)^{n \times k}$ and $\mathrm{Q}(\boldsymbol{\Omega})$ is the orthonormal matrix $\boldsymbol{Q}$ from the $QR$ decomposition of $\boldsymbol{\Omega}$. Let $\mu = \frac{n}{n-2}$ and variances $\sigma^2 = \frac{2n^2(n-1)}{(n-2)^2(n-4)}$. We can approximate $\mathbb{E}[\hat{\Delta}_0]^2$ and $\mathbb{E}[\Delta_0^{prior}]^2$ as follows:

$$\mathbb{E}[\Delta_l^{prior}]^2 \approx \left( \sqrt{2\sigma^2 \cdot \log(kn)\frac{k^2}{n^2}} + \frac{k\mu}{n} \right) \text{ and } \mathbb{E}[\hat{\Delta}_l]^2 \approx \left( \sqrt{2\sigma^2 \cdot \log(n)\frac{k}{n^2}} + \frac{k\mu}{n} \right). \quad (13)$$

**Empirical approximation of $r(k, n)$.** To compare the tightness of our proposed bound $\hat{\Delta}_0$ to the previous bound $\Delta_0^{prior}$, we perform both theoretical and empirical analyses at the first iteration, where the noise scaling depends only on the random initialization $\boldsymbol{X}^0$. Since $\boldsymbol{X}^0$ is independent of $\boldsymbol{A}$, we can estimate $\mathbb{E}[\Delta_0^{prior}]$ and $\mathbb{E}[\hat{\Delta}_0]$ by sampling a random Gaussian matrix $\boldsymbol{X}^0$.

*Experiment:* $r(k, n)$ depends on the number of factors $k$ and on $n$, where $(n \times n)$ is the size of $\boldsymbol{A}$. We therefore seek to compute it for multiple values of $n$ ($n \in \{8000, 12000\}$) and a range of values for $k$ (between 64

---

[6]Some work advocate reporting $\sigma^2$ instead (Balle and Wang, 2018). With fixed and using our CDP analysis, there is a one-to-one correspondence between the privacy budget $\epsilon$ and the noise variance $\sigma^2$. We opted to report $\epsilon$ because it conveys the level of privacy loss in our context (using composition) for the mechanism as a whole rather than raw noise variance at each step.

[7]We refer to Dwork et al. (2019) for more practical details on how to set $\epsilon$ and provide additional comparisons and details in Appendix B.1.

and 4000, with steps of 64), for each value of $n$. We use Approximation 6.2 to approximate it theoretically. To measure it empirically, we compute $\Delta_0^{prior}$ and $\hat{\Delta}_0$ based on $\boldsymbol{X}^0$ in the first step of the algorithm (it is independent of the matrix of interest $\boldsymbol{A}$), for $t$ runs of the algorithm. We can then use the $t$ measures to estimate $\mathbb{E}[\Delta_0^{prior}]$ and $\mathbb{E}[\hat{\Delta}_0]$. We find that $t = 5$ is enough to see a general trend, as we have estimates for many $(k, n)$ couples. We can also compare the empirical estimates to the proposed asymptotic approximations specified in Approximation 6.2.

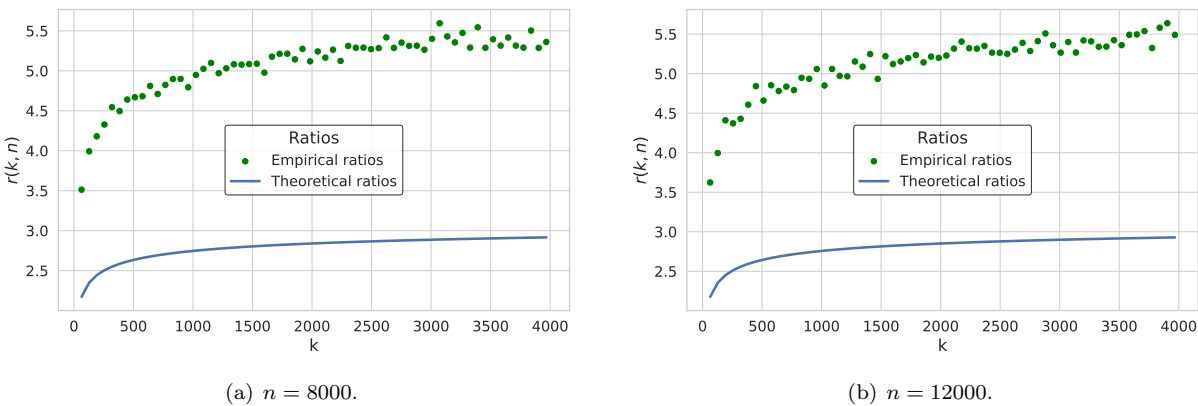

(a) $n = 8000$.          (b) $n = 12000$.

Figure 2: Comparison of empirically (green dots) and theoretically (blue line) estimated $r(k, n)$ ratios for $k$ ranging between 64 and 4000 with a step of 64. Empirical ratios are estimated using $t = 5$ runs of the first step of our algorithm, while theoretical ratios are based on Theorem 6.2. Results are shown for two different values of $n$: 8000 and 12000.

Figures 2(a) and 2(b) present the comparison between empirical (blue dots) and theoretical (green line) estimates of $r(k, n)$ for $n = 8000$ and $n = 12000$, respectively. The empirical ratios are calculated from the averages over the $t = 5$ runs for each value of $k$. The results indicate that both empirically and theoretically the proposed noise scaling $\hat{\Delta}_l$ is much tighter than $\Delta_l^{prior}$ at the first step of the algorithm. Our theoretical approximation is conservative and underestimates how much tighter the bound is initially, compared to what we observe in practice. The proposed bound is tighter by a multiplicative factor on the first step and therefore drastically reduces the impact of the noise introduced by DP at the first iteration. By noting that the power method is usually run for very few steps ($L$ is usually in the range of 1-5), this result complements our general convergence bounds on the overall algorithm derived in Theorems 3.3 and 4.1 and gives further intuition on the tightness and usefulness of our proposed sensitivity bound.

## 7 Conclusion

We presented Differentially Private versions of the centralized and decentralized randomized power method that addresses privacy concerns in large-scale spectral analysis and recommendation systems. We introduced a new sensitivity bound, which we show theoretically and empirically to improve the accuracy of the method while ensuring privacy guarantees. By employing Secure Aggregation in a decentralized setting, we can reduce the noise introduced for privacy, maintaining the efficiency and privacy of the centralized version but adapting it for distributed environments. Our methods could enable organizations (from healthcare networks to social platforms) to extract useful structure from data without compromising individual records.

**Limitations.** Our privacy guarantees require no dropout from participants to ensure distributed DP, which may not always be realistic. Indeed, if a client drops out, the variance of the aggregated noise is smaller than the expected variance, weakening the DP guarantee. Dropout-resilient strategies (e.g., oversampling, robust distributed DP) such as (Sabater et al., 2022) could be used. It would also be interesting to analyze a private and decentralized version of the accelerated version of the randomized power method, which could potentially converge faster.

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

# A  Proofs of Results

## A.1  Proof of Theorem 3.1

*Proof.* We denote by $\boldsymbol{A}'$ a matrix adjacent to $\boldsymbol{A}$ using (7):

$$\boldsymbol{A}' = \boldsymbol{A} + \boldsymbol{C}, \tag{14}$$

with $\boldsymbol{C}$ a symmetric matrix representing the update, subject to $\sqrt{\sum_i \|\boldsymbol{C}_{i:}\|_1^2} \leq 1$.

Then

$$
\begin{aligned}
\Delta_l &= \|\boldsymbol{A}\boldsymbol{X}^l - \boldsymbol{A}'\boldsymbol{X}^l\|_F \\
&= \|\boldsymbol{A}\boldsymbol{X}^l - (\boldsymbol{A} + \boldsymbol{C})\boldsymbol{X}^l\|_F \\
&= \|\boldsymbol{C}\boldsymbol{X}^l\|_F \\
&= \sqrt{\sum_i \|\boldsymbol{C}_{i:}\boldsymbol{X}^l\|_F^2}.
\end{aligned}
\tag{15}
$$

and

$$
\begin{aligned}
\|\boldsymbol{C}_{i:}\boldsymbol{X}^l\|_F &= \|\sum_j \boldsymbol{C}_{ij} \cdot \boldsymbol{e}_j^\top \cdot \boldsymbol{X}^l\|_F \\
&= \|\sum_j \boldsymbol{C}_{ij} \cdot \boldsymbol{X}_{j:}^l\|_F \\
&\leq \sum_j \|\boldsymbol{C}_{ij} \cdot \boldsymbol{X}_{j:}^l\|_F \\
&\leq \sum_j |\boldsymbol{C}_{ij}| \cdot \|\boldsymbol{X}_{j:}^l\|_F \\
&\leq \max_i \|\boldsymbol{X}_{i:}^l\|_F \cdot \sum_j |\boldsymbol{C}_{ij}|.
\end{aligned}
\tag{16}
$$

By injecting (15) into (16), we have:

$$
\begin{aligned}
\Delta_l &= \sqrt{\sum_i \|\boldsymbol{C}_{i:}\boldsymbol{X}^l\|_F^2} \\
&\leq \sqrt{\sum_i (\max_i \|\boldsymbol{X}_{i:}^l\|_F \cdot \sum_j |\boldsymbol{C}_{ij}|)^2} \\
&\leq \max_i \|\boldsymbol{X}_{i:}^l\|_F \sqrt{\sum_i (\cdot \sum_j |\boldsymbol{C}_{ij}|)^2} \\
&\leq \max_i \|\boldsymbol{X}_{i:}^l\|_F \sqrt{\sum_i \|\boldsymbol{C}_{i:}\|_1^2} \\
&\leq \max_i \|\boldsymbol{X}_{i:}^l\|_F \triangleq \hat{\Delta}_l.
\end{aligned}
\tag{17}
$$

Although the proposed update model (7) is more general than when $\boldsymbol{A}'$ is defined using (6), the proposed bound $\hat{\Delta}_l$ is also generally tighter than the bound proposed in Hardt and Price (2014); Balcan et al. (2016).

$\square$

## A.2 Privacy proof

### A.2.1 Note on related privacy proofs:

Several Differential Privacy (DP) proofs for the private randomized power method (PPM) have been developed in prior works, for instance those by Hardt and Roth (2012; 2013); Hardt and Price (2014); Balcan et al. (2016). These papers mostly rely on the same proofs to establish that the PPM satisfies $(\epsilon, \delta)$-DP. Specifically, Balcan et al. (2016) references the privacy proof from Hardt and Price (2014), which builds upon the privacy proof from Hardt and Roth (2013). The proof therein is also closely related to the one of Hardt and Roth (2012).

However, both Hardt and Roth (2013) (Theorem 2.4) and Hardt and Roth (2012) (Theorem 2.4) contain errors in their proposed composition rules, where a comparison sign is mistakenly flipped. This error could potentially cause the privacy parameter $\epsilon$ in the proposed mechanism to be arbitrarily small, providing no privacy guarantee at all. The original composition rule is presented in Theorem III.3 of Dwork et al. (2010). Moreover, Lemma 3.4 of Hardt and Roth (2013) misuses Theorem 2.4. Indeed, they claim that their algorithm satisfies $(\epsilon', \delta)$-DP at each iteration. By their Theorem 2.4, then the algorithm overall should satisfy $(\epsilon', k\delta + \delta')$-DP where $\delta' > 0$, and not $(\epsilon', \delta)$-DP as claimed.

### A.2.2 Zero-Concentrated Differential Privacy (zCDP):

A randomized algorithm $\mathcal{M}$ is said to satisfy $\rho$-zero-Concentrated Differential Privacy (zCDP) if for all neighboring datasets $D_1$ and $D_2$, and for all $\alpha \in (1, \infty)$, the following holds:

$$D_\alpha(\mathcal{M}(D_1) \| \mathcal{M}(D_2)) \leq \rho\alpha, \tag{18}$$

where $D_\alpha$ is the Rényi divergence of order $\alpha$ and $\rho$ is a positive parameter controlling the trade-off between privacy and accuracy (smaller values of $\rho$ imply stronger privacy guarantees).

The following lemma, introduced by Bun and Steinke (2016), specifies how the addition of Gaussian noise can be used to construct a randomized algorithm that satisfies zCDP.

**Lemma A.1** (Gaussian Mechanism (Proposition 1.6 (Bun and Steinke, 2016))). *Let $f : X^n \to \mathbb{R}$ be a sensitivity-$\Delta$ function. Consider the mechanism (randomized algorithm) $M : X^n \to \mathbb{R}$, defined as $M(x) = f(x) + Z_x$ where for each input $x$, $Z_x$ is independently drawn from $\mathcal{N}(0, \sigma^2)$. Then $M$ satisfies $\left(\frac{\Delta^2}{2\sigma^2}\right)$-zCDP.*

The next lemma, which is a generalized version of a result presented by Bun and Steinke (2016), explains how a randomized algorithm, constructed by recursively composing a sequence of zCDP-satisying randomized algorithms, also satisfies zCDP.

**Lemma A.2** (Adaptive composition (Generalization from Lemma 2.3 of (Bun and Steinke, 2016))). *Let $M_1 : X^n \to Y_1$, $M_2 : X^n \times Y_1 \to Y_2$, $\ldots$, $M_L : X^n \times Y_1 \times \cdots \times Y_{L-1} \to Y_L$ be randomized algorithms. Suppose $M_i$ satisfies $\rho_i$-zCDP as a function of its first argument for each $i = 1, 2, \ldots, L$. Let $M'' : X^n \to Y_L$, constructed recursively by:*

$$M''(x) = M_L(x, M_{L-1}(x, \ldots, M_2(x, M_1(x)) \ldots)). \tag{19}$$

*Then $M''$ satisfies $(\sum_{i=1}^{L} \rho_i)$-zCDP.*

### A.2.3 Proof of Theorem 3.2

*Proof.* We can see lines 4-5 of Algorithm 1 as a sequential composition ($M$) of $L$ Gaussian Mechanisms. By Lemma A.1, each mechanism $M_i$ satisfies $(\frac{\Delta_i^2}{2\sigma_i^2})$-zCDP where $\Delta_i$ is the $\ell_2$-sensitivity of the function associated to mechanism $\Delta_i$ and $\sigma_i^2$ is the variance of the noise added with the Gaussian Mechanism. By Lemma A.2, the composition of mechanisms $M = (M_1, ..., M_i, ..., M_L)$ satisfies $(\sum_{i=1}^{L} \frac{\Delta_i^2}{2\sigma_i^2})$-zCDP. Let $\rho \triangleq \sum_{i=1}^{L} \frac{\Delta_i^2}{2\sigma_i^2}$. By

design of our algorithm, we have:

$$\rho = \sum_{i=1}^{L} \frac{\Delta_i^2}{2\sigma_i^2} \tag{20}$$

$$\leq \sum_{i=1}^{L} \frac{(\hat{\Delta}_l)^2}{2\sigma_i^2} \tag{21}$$

$$= \sum_{i=1}^{L} \frac{1}{2\sigma^2} \tag{22}$$

$$= \sum_{i=1}^{L} \frac{\epsilon^2}{8L \log(1/\delta)} \tag{23}$$

$$= \frac{\epsilon^2}{8 \log(1/\delta)}. \tag{24}$$

By Proposition 1.3 of Bun and Steinke (2016), if M provides $\rho$-zCDP, then M is $(\rho + 2\sqrt{\rho \log(1/\delta)}, \delta)$-DP, $\forall \delta > 0$. Let $\epsilon' \triangleq \rho + 2\sqrt{\rho \log(1/\delta)}$. Then:

$$\epsilon' = \rho + 2\sqrt{\rho \log(1/\delta)} \tag{25}$$

$$\leq \frac{\epsilon^2}{8 \log(1/\delta)} + 2\sqrt{\frac{\epsilon^2}{8 \log(1/\delta)} \cdot \log(1/\delta)} \tag{26}$$

$$\leq \frac{\epsilon^2}{8 \log(1/\delta)} + \frac{\epsilon}{2}. \tag{27}$$

$$\tag{28}$$

To satisfy $(\epsilon, \delta)$-DP, we need:

$$\epsilon \geq \epsilon' \impliedby \epsilon \geq \frac{\epsilon^2}{8 \log(1/\delta)} + \frac{\epsilon}{2} \tag{29}$$

$$\iff \frac{\epsilon}{2} \geq \frac{\epsilon^2}{8 \log(1/\delta)} \tag{30}$$

$$\iff \epsilon \leq 4 \log(1/\delta) \tag{31}$$

$$\iff \delta \leq \exp\left(-\frac{\epsilon}{4}\right), \tag{32}$$

which is a reasonable assumption, as in practice $\epsilon = O(1)$ and $\delta \ll \frac{1}{d}$, where $d$ is the number of records to protect. In our case, $d = n^2$ because we run the privacy-preserving power method on $\boldsymbol{A} \in \mathbb{R}^{n \times n}$. $\qquad \square$

### A.3 Proof of Theorem 3.3

The following theorem from Balcan et al. (2016) will be useful in our proof:

**Theorem A.3** (Bound for the noisy power method (NPM) (Balcan et al., 2016))**.** *Let $k \leq q \leq p$ be positive integers. Let $\boldsymbol{U}_q \in \mathbb{R}^{d \times q}$ be the top-$q$ eigenvectors of a positive semi-definite matrix $\boldsymbol{A}$ and let $\lambda_1 \geq \cdots \geq \lambda_d \geq 0$ denote its eigenvalues and fix $\eta = O\left(\frac{\lambda_q}{\lambda_k} \cdot \min\left\{\frac{1}{\log\left(\frac{\lambda_k}{\lambda_q}\right)}, \frac{1}{\log(\tau d)}\right\}\right)$. If at every iteration $l$ of the NPM $\boldsymbol{G}_\ell$ satisfies Assumption 2.1, then after*

$$L = \Theta\left(\frac{\lambda_k}{\lambda_k - \lambda_{q+1}} \log\left(\frac{\tau d}{\eta}\right)\right).$$

*iterations, with probability at least $1 - \tau^{-\Omega(p+1-q)} - e^{-\Omega(d)}$, we have:*

$$\|(\boldsymbol{I} - \boldsymbol{X}^L \boldsymbol{X}^{L^\top})\boldsymbol{U}_k\|_2 \leq \eta \quad \text{and} \quad \|(\boldsymbol{I} - \boldsymbol{X}^L(\boldsymbol{X}^L)^\top)\boldsymbol{A}\|_2^2 \leq \lambda_{k+1}^2 + \eta^2 \lambda_k^2. \tag{33}$$

We now provide the proof of Theorem 3.3:

*Proof.* According to Hardt and Price (2014), if $\boldsymbol{G}_l \sim \mathcal{N}(0, \sigma_l)^{d \times p}$, then with probability 99/100 we have:

$$\max_{l=1}^{L} \|\boldsymbol{G}_l\| \leq \sigma_l \cdot \sqrt{d \cdot \log(L)}\,,$$
$$\max_{l=1}^{L} \|\boldsymbol{U}^\top \boldsymbol{G}_l\| \leq \sigma_l \cdot \sqrt{p \cdot \log(L)}\,. \tag{34}$$

We can therefore satisfy the noise conditions of Theorem A.3 with probability 99/100 if we choose $\eta = \frac{\sigma_l \cdot \sqrt{d \cdot \log(L)}}{\lambda_k - \lambda_{q+1}}$, giving us:

$$\|(\boldsymbol{I} - \boldsymbol{X}^L {\boldsymbol{X}^L}^\top) \cdot \boldsymbol{U}_k\| \leq \frac{\sigma_l \cdot \sqrt{d \cdot \log(L)}}{\lambda_k - \lambda_{q+1}}\,, \tag{35}$$

which leads us to the statement of the theorem by injecting $\sigma_l = \hat{\Delta}_l \cdot \epsilon^{-1} \sqrt{4L \log(1/\delta)}$. $\qquad\square$

## A.4 Proof of Theorem 4.1

We provide here the proof for Theorem 4.1:

**Theorem 4.1. *Improved PPM with Runtime-Independent Bound.*** *Let $\boldsymbol{A} \in \mathbb{R}^{n \times n}$ be a symmetric data matrix. Fix target rank $k$, intermediate rank $q \geq k$ and iteration rank $p$ with $2q \leq p \leq n$. Suppose the number of iterations $L$ is set as $L = \Theta(\frac{\lambda_k}{\lambda_k - \lambda_{q+1}} \log(n))$. Let $\boldsymbol{U}_q \in \mathbb{R}^{n \times q}$ be the top-$q$ eigenvectors of $\boldsymbol{A}$ and let $\lambda_1 \geq \cdots \geq \lambda_n \geq 0$ denote its eigenvalues. Let $\delta \in (0, 1)$ and $\epsilon > 0$ be privacy parameters such that $\delta \leq \exp\left(-\frac{\epsilon}{4}\right)$. Then Algorithm 1 is ($\epsilon$,$\delta$)-DP and we have with probability at least 0.9*

$$\|(\boldsymbol{I} - \boldsymbol{X}^L (\boldsymbol{X}^L)^\top) \boldsymbol{U}_k\|_2 \leq \eta \quad and \quad \|(\boldsymbol{I} - \boldsymbol{X}^L (\boldsymbol{X}^L)^\top) \boldsymbol{A}\|_2^2 \leq \lambda_{k+1}^2 + \eta^2 \lambda_k^2 \tag{11}$$

$$with \quad \eta = O\left( \frac{\epsilon^{-1} \cdot \min(\mu_0(\boldsymbol{A}) \sqrt{p \cdot \log(n)}, \mu_1(\boldsymbol{A})) \cdot \sqrt{Ln \log(1/\delta) \log(L)}}{\lambda_k - \lambda_{q+1}} \right). \tag{12}$$

*Proof.* We showed in Equation that $\Delta_l \leq \max_i \|\boldsymbol{X}_{i:}^l\|_F$. This quantity depends on values computed during the execution of the algorithm. We now show that we can bound $\max_i \|\boldsymbol{X}_{i:}^l\|_F$ with a runtime-independent bound.

Without loss of generality and to simplify notation, we use $\boldsymbol{X}$ to denote any matrix $\boldsymbol{X}_l$ computed during the execution of the Private Power Method. Let $\boldsymbol{X}_{:c} = x_c$ denote a column of $\boldsymbol{X}$.

As $\boldsymbol{A}$ is an Hermitian matrix, by the spectral theorem, we have $\boldsymbol{A} = \boldsymbol{U} \boldsymbol{D} \boldsymbol{U}^\top$, where $\boldsymbol{U}$ is unitary (with orthonormal columns) and $\boldsymbol{D}$ is diagonal.

As the columns of $\boldsymbol{U}$ form a complete basis for $\mathbb{R}^n$, we can write any column $x_c$ of $\boldsymbol{X}$ as $\sum_{i=1}^{n} \alpha_c^i u_i$, where $u_i$ denotes the $i$-th eigenvector of $\boldsymbol{A}$, and $\alpha_c^i$ is a scalar.

We can then write:

$$\langle x_c, x_e \rangle = \langle \sum_{i=1}^{n} \alpha_c^i u_i, \sum_{j=1}^{n} \alpha_e^j u_j \rangle \tag{36}$$

$$= \sum_{i=1}^{n} \sum_{j=1}^{n} \alpha_c^i \alpha_e^j \langle u_i, u_j \rangle \tag{37}$$

$$= \sum_{i=1}^{n} \alpha_c^i \alpha_e^i. \qquad \text{(Orth. columns of } \boldsymbol{U}) \tag{38}$$

$\boldsymbol{X}$ is the "$Q$" matrix constructed from a Gram-Schmidt QR decomposition, it has therefore orthonormal columns by definition. Therefore, if $c = e$, then $\langle x_c, x_e \rangle = \sum_{i=1}^{n} (\alpha_c^i)^2 = 1$. Otherwise, we have $\langle x_c, x_e \rangle = \sum_{i=1}^{n} \alpha_c^i \alpha_e^i = 0$.

The key is then to notice that we can define a matrix $\boldsymbol{B} \in \mathbb{R}^{n \times p}$ with orthonormal columns such that $\boldsymbol{X} = \boldsymbol{U}\boldsymbol{B}$ and $\boldsymbol{B}_{jc} = \alpha_c^j$.

We recall that any matrices $\boldsymbol{H}$ and $\boldsymbol{J}$, we have $\|\boldsymbol{H}\boldsymbol{J}\|_F \leq \|\boldsymbol{H}\|_2 \|\boldsymbol{J}\|_F$.

Then, we can bound the norm of any row of $\boldsymbol{X}_{i:}$ as:

$$\|\boldsymbol{X}_{i:}\|_F = \|\boldsymbol{X}_{i:}^\top\|_F, \tag{39}$$

$$= \|\boldsymbol{B}^\top \boldsymbol{U}_{i:}^\top\|_F, \tag{40}$$

$$\leq \|\boldsymbol{B}^\top\|_2 \|\boldsymbol{U}_{i:}^\top\|_F. \tag{41}$$

$\boldsymbol{B}$ has orthonormal columns by construction, therefore

$$\|\boldsymbol{B}\|_2 = \sqrt{\|\boldsymbol{B}^\top \boldsymbol{B}\|_2} \tag{42}$$

$$= \sqrt{\|\boldsymbol{I}\|_2} \tag{43}$$

$$= 1. \tag{44}$$

We then have:

$$\|\boldsymbol{X}_{i:}\|_F \leq \|\boldsymbol{B}^\top\|_2 \|\boldsymbol{U}_{i:}^\top\|_F \tag{45}$$

$$\leq \|\boldsymbol{U}_{i:}^\top\|_F \tag{46}$$

$$\leq \|\boldsymbol{U}_{i:}\|_F. \tag{47}$$

Additionally as $\boldsymbol{X}_{i:} \in \mathbb{R}^{1 \times n}$, $\max_i \|\boldsymbol{X}_{i:}\|_2 = \max_i \|\boldsymbol{X}_{i:}\|_F \leq \max_i (\|\boldsymbol{U}_{i:}\|_F) = \mu_1(\boldsymbol{A})$.

**Note:** By Section 2.4 from Woodruff (2014), for any row $\boldsymbol{v}$ of a matrix with orthonormal columns $\boldsymbol{Z}$, $\|v\|_2 \leq 1$.

As $\boldsymbol{U}$ has orthonormal columns by construction, $\max_i \|\boldsymbol{U}_{i:}\|_2 \leq 1$.

We can therefore bound $\max_i \|\boldsymbol{X}_{i:}\|_2$ as:

$$\Delta_l \leq \max_i \|\boldsymbol{X}_{i:}\|_2 \leq \mu_1(\boldsymbol{A}) \leq 1. \tag{48}$$

Injecting this bound in Theorem 3.3 leads us to Theorem 4.1, giving us a runtime-independent bound. $\qquad\square$

### A.5 Proof of Theorem 5.2

*Proof.* It is straightforward to see that if steps 2 and 3 of Algorithm 2 are equivalent to step 1 of Algorithm 1, then the two algorithms are equivalent. Recall that $Y_\ell^{(i)} = \boldsymbol{A}^{(i)} \boldsymbol{X}^{\ell-1} + \boldsymbol{G}_\ell^{(i)}$ and $\boldsymbol{G}_\ell^{(i)} \sim \mathcal{N}(0, \Delta_l^2 \nu^2)^{n \times p}$. Then steps 2 and 3 of Algorithm 2 correspond to:

$$\boldsymbol{Y}_\ell = SecAgg(\boldsymbol{Y}_\ell^{(i)}, \{i | 1 \leq i \leq s\})$$

$$= \sum_{i=1}^{s} \boldsymbol{Y}_\ell^{(i)}$$

$$= \sum_{i=1}^{s} (\boldsymbol{A}^{(i)} \boldsymbol{X}_{\ell-1} + \boldsymbol{G}_\ell^{(i)}) \tag{49}$$

$$= \boldsymbol{A}\boldsymbol{X}^{\ell-1} + \sum_{i=1}^{s} \boldsymbol{G}_\ell^{(i)}$$

$$= \boldsymbol{A}\boldsymbol{X}^{\ell-1} + \boldsymbol{G}_\ell,$$

where $\boldsymbol{G}_\ell \sim \mathcal{N}(0, \Delta_l^2 \cdot (s\nu^2))^{n \times p}$, and we have $s\nu^2 = \sigma^2$ by definition, completing the equivalence proof. $\quad\square$

## B   Application to recommender systems:

Let $s$ be the number of users in our system and $n$ the number of items. Let $\boldsymbol{R} \in \mathbb{R}^{s \times n}$ be the user-item interaction matrix, such that $\boldsymbol{R}_{ui} = 1$ only if user $u$ has interacted with item $i$, and $\boldsymbol{R}_{ui} = 0$ else. Let $\boldsymbol{U} = \mathrm{Diag}(\boldsymbol{R} \cdot \mathbf{1}_{|\mathcal{I}|})$ be the user degrees matrix, and $\boldsymbol{I} = \mathrm{Diag}(\mathbf{1}_{|\mathcal{I}|}^\top \cdot \boldsymbol{R})$ the item degrees matrix. In Shen et al. (2021); Choi et al. (2023), the normalized interaction matrix is defined as:

$$\boldsymbol{R}' = \boldsymbol{U}^{-\frac{1}{2}} \boldsymbol{R} \boldsymbol{I}^{-\frac{1}{2}},$$

and the item-item normalized adjacency matrix as:

$$\begin{aligned}
\boldsymbol{P}' &= \tilde{\boldsymbol{R}}^\top \tilde{\boldsymbol{R}} \\
&= (\boldsymbol{U}^{-\frac{1}{2}} \boldsymbol{R})^\top (\boldsymbol{U}^{-\frac{1}{2}} \boldsymbol{R}) \\
&= (\boldsymbol{U}^{-\frac{1}{2}} \boldsymbol{R} \boldsymbol{I}^{-\frac{1}{2}})^\top (\boldsymbol{U}^{-\frac{1}{2}} \boldsymbol{R} \boldsymbol{I}^{-\frac{1}{2}}).
\end{aligned} \tag{50}$$

To simplify our analysis, we consider $\boldsymbol{I}$ public and do not use item-wise normalization in the computation of the ideal low-pass filter, leaving it for future work. We therefore define $\tilde{\boldsymbol{R}} = \boldsymbol{U}^{-\frac{1}{2}} \boldsymbol{R}$ and $\tilde{\boldsymbol{P}} = (\boldsymbol{U}^{-\frac{1}{2}} \boldsymbol{R})^\top (\boldsymbol{U}^{-\frac{1}{2}} \boldsymbol{R})$.

**Lemma B.1.** *We can use Algorithm 2 with $\Delta_l = \sqrt{2} \max_i \|\boldsymbol{X}_{i:}^l\|_2$ to compute the top-p eigenvectors of $\tilde{\boldsymbol{P}}$ in a decentralized setting with under a $(\epsilon, \delta)$-Differential Privacy guarantee.*

*Proof.* Let $\tilde{\boldsymbol{P}}_{ij}$ denote the element of $\tilde{\boldsymbol{P}}$ at row $i$ and column $j$ and let $d_{user}(u) = \sum_{i=0}^{n-1} r_{ui}$. We can write $\tilde{\boldsymbol{P}}_{ij}$ as:

$$\begin{aligned}
\tilde{\boldsymbol{P}}_{ij} &= ((\boldsymbol{U}^{-\frac{1}{2}} \boldsymbol{R})^\top (\boldsymbol{U}^{-\frac{1}{2}} \boldsymbol{R}))_{ij} \\
&= (\boldsymbol{U}^{-\frac{1}{2}} \boldsymbol{R})^\top)_{i,*} (\boldsymbol{U}^{-\frac{1}{2}} \boldsymbol{R})_{*,j} \\
&= ((\boldsymbol{U}^{-\frac{1}{2}} \boldsymbol{R})_{*,i})^\top (\boldsymbol{U}^{-\frac{1}{2}} \boldsymbol{R})_{*,j} \\
&= \sum_{u=0}^{s-1} \frac{1}{\sqrt{d_{user}(u)}} r_{ui} \frac{1}{\sqrt{d_{user}(u)}} r_{uj} \\
&= \sum_{u=0}^{s-1} \frac{1}{d_{user}(u)} \cdot r_{ui} \cdot r_{uj}.
\end{aligned} \tag{51}$$

By noticing that $(\boldsymbol{R}^\top \boldsymbol{R})_{ij} = \sum_u r_{ui} \cdot r_{uj}$, we can deduce that $\tilde{\boldsymbol{P}} = \sum_u \frac{1}{d_{user}(u)} \boldsymbol{R}_u^\top \boldsymbol{R}_u$. Therefore $\tilde{\boldsymbol{P}}$ is partitioned among $s$ users as described in Section 5.

Sensitivity: We protect the user at the item-level and use the deletion model of differential privacy to compute the sensitivity of the PPM used with the item-item normalized adjacency matrix ($\boldsymbol{A} = \tilde{\boldsymbol{P}}$). Therefore we have:

$$\boldsymbol{A}_{ij} = \sum_{u=0}^{s-1} \frac{1}{d_{user}(u)} \cdot r_{ui} \cdot r_{uj}, \tag{52}$$

and

$$\boldsymbol{A}'_{ij} = \sum_{u=0}^{s-1} \frac{1}{d_{user}(u) - 1} \cdot r'_{ui} \cdot r'_{uj}, \tag{53}$$

where $r'_{ui} = r_{ui}$ except for one user-item interaction, *i.e.*, $r_{vk} = 1$ but $r'_{vk} = 0$. Let $\boldsymbol{C} = \boldsymbol{A} - \boldsymbol{A}'$. Let $\mathcal{N}(v)$ be the set of items which user $v$ liked before deletion. We have:

$$\sum_i \|\boldsymbol{C}_{i:}\|_1^2 = \sum_i (\sum_j |\sum_{u=0}^{s-1} \frac{1}{d_{user}(u)} \cdot r_{ui} \cdot r_{uj} - \sum_{u=0}^{s-1} \frac{1}{d_{user}(u)-1} \cdot r'_{ui} \cdot r'_{uj}|)^2 \tag{54}$$

$$= \sum_i (\sum_j |\frac{1}{d_{user}(v)} \cdot r_{vi} \cdot r_{vj} - \frac{1}{d_{user}(v)-1} \cdot r'_{vi} \cdot r'_{vj}|)^2 \tag{55}$$

$$= \sum_{i \in \mathcal{N}(v)} (\sum_{j \in \mathcal{N}(v)} |\frac{1}{d_{user}(v)} \cdot r_{vi} \cdot r_{vj} - \frac{1}{d_{user}(v)-1} \cdot r'_{vi} \cdot r'_{vj}|)^2 \tag{56}$$

$$= \sum_{i \in \{\mathcal{N}(v) \setminus k\}} (\sum_{j \in \mathcal{N}(v)} |\frac{1}{d_{user}(v)} \cdot r_{vi} \cdot r_{vj} - \frac{1}{d_{user}(v)-1} \cdot r'_{vi} \cdot r'_{vj}|)^2 \tag{57}$$

$$+ (\sum_{j \in \mathcal{N}(v)} |\frac{1}{d_{user}(v)} \cdot r_{vk} \cdot r_{vj} - \frac{1}{d_{user}(v)-1} \cdot r'_{vk} \cdot r'_{vj}|)^2. \tag{58}$$

We have:

$$\sum_{i \in \{\mathcal{N}(v) \setminus k\}} (\sum_{j \in \mathcal{N}(v)} |\frac{1}{d_{user}(v)} \cdot r_{vi} \cdot r_{vj} - \frac{1}{d_{user}(v)-1} \cdot r'_{vi} \cdot r'_{vj}|)^2 \tag{59}$$

$$= \sum_{i \in \{\mathcal{N}(v) \setminus k\}} (\sum_{j \in \{\mathcal{N}(v) \setminus k\}} |\frac{1}{d_{user}(v)} \cdot r_{vi} \cdot r_{vj} - \frac{1}{d_{user}(v)-1} \cdot r'_{vi} \cdot r'_{vj}| + \frac{1}{d_{user}(v)} \cdot r_{vi} \cdot r_{vk})^2 \tag{60}$$

$$= (d_{user}(v) - 1)\{(d_{user}(v)-1)|\frac{1}{d_{user}(v)} - \frac{1}{d_{user}(v)-1}| + \frac{1}{d_{user}(v)}\}^2 \tag{61}$$

$$= (d_{user}(v) - 1)\{|\frac{1}{d_{user}(v)}| + \frac{1}{d_{user}(v)}\}^2 \tag{62}$$

$$= \frac{2(d_{user}(v) - 1)}{d_{user}(v)^2}, \tag{63}$$

and

$$(\sum_{j \in \mathcal{N}(v)} |\frac{1}{d_{user}(v)} \cdot r_{vk} \cdot r_{vj} - \frac{1}{d_{user}(v)-1} \cdot r'_{vk} \cdot r'_{vj}|)^2 \tag{64}$$

$$= (\sum_{j \in \mathcal{N}(v)} |\frac{1}{d_{user}(v)} \cdot r_{vj}|)^2 \tag{65}$$

$$= (d_{user}(v)|\frac{1}{d_{user}(v)}|)^2 \tag{66}$$

$$= 1. \tag{67}$$

By noticing that $\frac{2(d_{user}(v)-1)}{d_{user}(v)^2} \leq 1$, we can deduce that:

$$\sum_i \|\boldsymbol{C}_{i:}\|_1^2 = \sum_{i \in \{\mathcal{N}(v) \setminus k\}} (\sum_{j \in \mathcal{N}(v)} |\frac{1}{d_{user}(v)} \cdot r_{vi} \cdot r_{vj} - \frac{1}{d_{user}(v)-1} \cdot r'_{vi} \cdot r'_{vj}|)^2 \tag{68}$$

$$+ (\sum_{j \in \mathcal{N}(v)} |\frac{1}{d_{user}(v)} \cdot r_{vk} \cdot r_{vj} - \frac{1}{d_{user}(v)-1} \cdot r'_{vk} \cdot r'_{vj}|)^2 \tag{69}$$

$$= \frac{2(d_{user}(v) - 1)}{d_{user}(v)^2} + 1 \tag{70}$$

$$\leq 2 \tag{71}$$

$$\implies \sqrt{\sum_i \|\boldsymbol{C}_{i:}\|_1^2} \leq \sqrt{2}. \tag{72}$$

By Equation 17,

$$\Delta_l \leq \max_i \|\boldsymbol{X}_{i:}^l\|_F \sqrt{\sum_i \|\boldsymbol{C}_{i:}\|_1^2} \tag{73}$$

$$\implies \Delta_l \leq \max_i \|\boldsymbol{X}_{i:}^l\|_F \sqrt{2}. \tag{74}$$

$$\square$$

### B.1 Approximation errors comparisons:

As explained in Section 6.1, GF-CF and BSPM (with its parameter $T_b = 1$) use $\boldsymbol{U}_p$ to compute the ideal low-pass filter and filter the interaction matrix $\boldsymbol{R}$, yielding $\boldsymbol{R}_p$. Indeed, we have:

$$\boldsymbol{R}_p = \boldsymbol{R} \cdot \boldsymbol{I}^{-\frac{1}{2}} \boldsymbol{U}_p \boldsymbol{U}_p^\top \boldsymbol{I}^{\frac{1}{2}}. \tag{75}$$

To study the impact of Differential Privacy on our system, we compute approximations of $\boldsymbol{U}_p$ (denoted by $\tilde{\boldsymbol{U}}_p$) using our proposed decentralized PPM, the PPM versions of Balcan et al. (2016) or FedPower from Guo et al. (2024). We have:

$$\tilde{\boldsymbol{R}}_p = \boldsymbol{R} \cdot \boldsymbol{I}^{-\frac{1}{2}} \tilde{\boldsymbol{U}}_p \tilde{\boldsymbol{U}}_p^\top \boldsymbol{I}^{\frac{1}{2}}. \tag{76}$$

We then define the relative approximation error caused by the use of differential privacy as $\frac{\|\tilde{\boldsymbol{R}}_p - \boldsymbol{R}_p\|_F}{\|\boldsymbol{R}_p\|_F}$.

We use $L = 3$ to run the Power Method as it is the default hyper-parameter choice from Shen et al. (2021); Choi et al. (2023). We use $p = 32$ to have acceptable approximation error for reasonable values of $\epsilon$ (5-10). We use the synchronous version of FedPower (Guo et al., 2024) to simplify the comparison, *i.e.*, we set their parameter $\mathcal{I}_T = L$. We note that FedPower could be improved by also using Secure Aggregation and therefore reducing the noise necessary for DP. It might also benefit from our sensitivity analysis in the synchronous setting.

We showed the impact of the Differential Privacy parameter $\epsilon$ on the approximation errors for the MovieLens and EachMovie datasets in Figures 1(a) and 1(b).The trends for the approximation errors on the Jester dataset are quite similar, as shown in Figure 3. We however note that all methods perform better on this dataset. We hypothesize that this is because the Jester dataset is much more dense compared to the EachMovie and MovieLens datasets, hence the ratio of magnitude of the noise added due to DP compared to the magnitude of the elements of the item-item matrix is smaller on this dataset.

### B.2 Experimental Details

**Datasets.** We benchmark our method on four standard recommendation datasets: MovieLens-10M, EachMovie, Jester, and Amazon-book. For Amazon-book, we use the publicly available 80%/20% train/test splits; for EachMovie, MovieLens-10M and Jester we use the same train/test proportions and therefore apply a 80%/20% split at the user level (each user retains at least one test interaction). Table 3 summarizes the post-processing statistics.

Table 3: Dataset statistics after preprocessing: number of users, items, interactions.

| Dataset | Users | Items | Interactions |
|---|---|---|---|
| MovieLens-10M | 71,567 | 10,677 | 7,972,582 |
| EachMovie | 74,425 | 1,649 | 2,216,887 |
| Jester | 54,906 | 151 | 1,450,010 |
| Amazon-book | 52,643 | 91,599 | 2,984,108 |

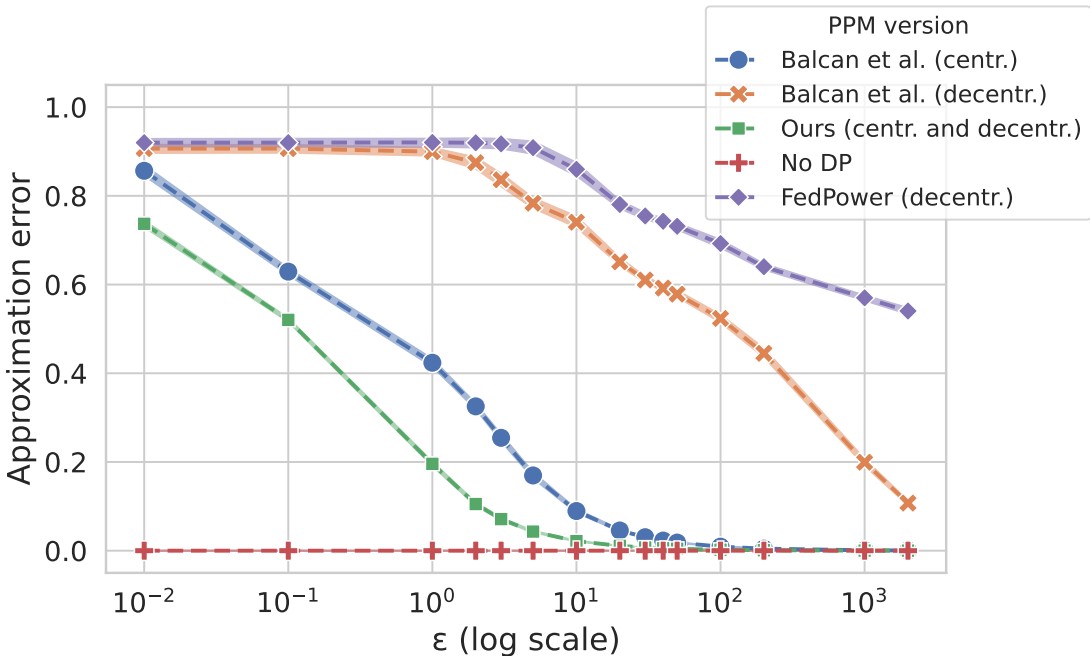

Figure 3: Impact of the Differential Privacy parameter $\epsilon$ on the relative approximation error $\frac{\|\tilde{\boldsymbol{R}}_p - \boldsymbol{R}_p\|_F}{\|\boldsymbol{R}_p\|_F}$ associated to multiple PPM versions, computed on 10 runs with 99% confidence intervals (computed via bootstrap), with $p = 32$, $L = 3$ and $\delta = 10^{-4}$ on the Jester dataset.

**Hyperparameters.** We fix the number of power iterations to $L = 3$ to correspond to PyTorch's `svd_lowrank` approximate basis subroutine (Algorithm 4.3 in (Halko et al., 2011)). No further hyperparameter tuning is performed.

**Experimental Protocol and Runs.** Each "run" uses a different Pseudo Random Number Generator seed for the Gaussian initialization $\boldsymbol{G}_0$. We perform 10 runs per setting (for one given $\epsilon$ value, all methods are initialized with the same Gaussian matrix), varying only the PRNG seed across runs.

**Confidence Intervals.** We report the mean and a 99% confidence interval over the 10 runs, estimated via nonparametric bootstrap (1 000 resamples).

**Computational Resources.** All experiments ran on a 16-core CPU. Figure 1 (10 runs × 14 '$\epsilon$' values × 5 methods) requires $\approx$ 30 min of wall-clock time.

### B.3 Impact of the number of components

Figure 4 and Figure 5 report the relative approximation error as a function of the privacy parameter $\epsilon$ for different numbers of extracted components $p$ on EachMovie and Jester. On the EachMovie dataset, we find that for small to medium numbers of components (e.g., $p = 4$ or $p = 16$), our decentralized private power method consistently achieves lower approximation error than all competing methods, including the centralized Analyze Gauss baseline. However, as the number of components increases (e.g., $p = 64$ or $p = 128$), only Analyze Gauss eventually surpasses our decentralized method in terms of approximation accuracy.

On the Jester dataset (considerably smaller), we observe the same patterns except that the centralized Analyze Gauss baseline performs on par with our decentralized method across the range of tested component numbers. This behavior suggests that in very small-scale problems, the centralized mechanism may still

provide a competitive or even superior balance of accuracy and privacy, despite its impractical communication and memory costs in a decentralized environment.

Finally, we note that it was not possible to run Analyze Gauss on the MovieLens-10m dataset due to prohibitive memory requirements, which further highlights its practical limitations in large-scale settings. In contrast, our decentralized method still achieves favorable privacy-utility trade-offs.

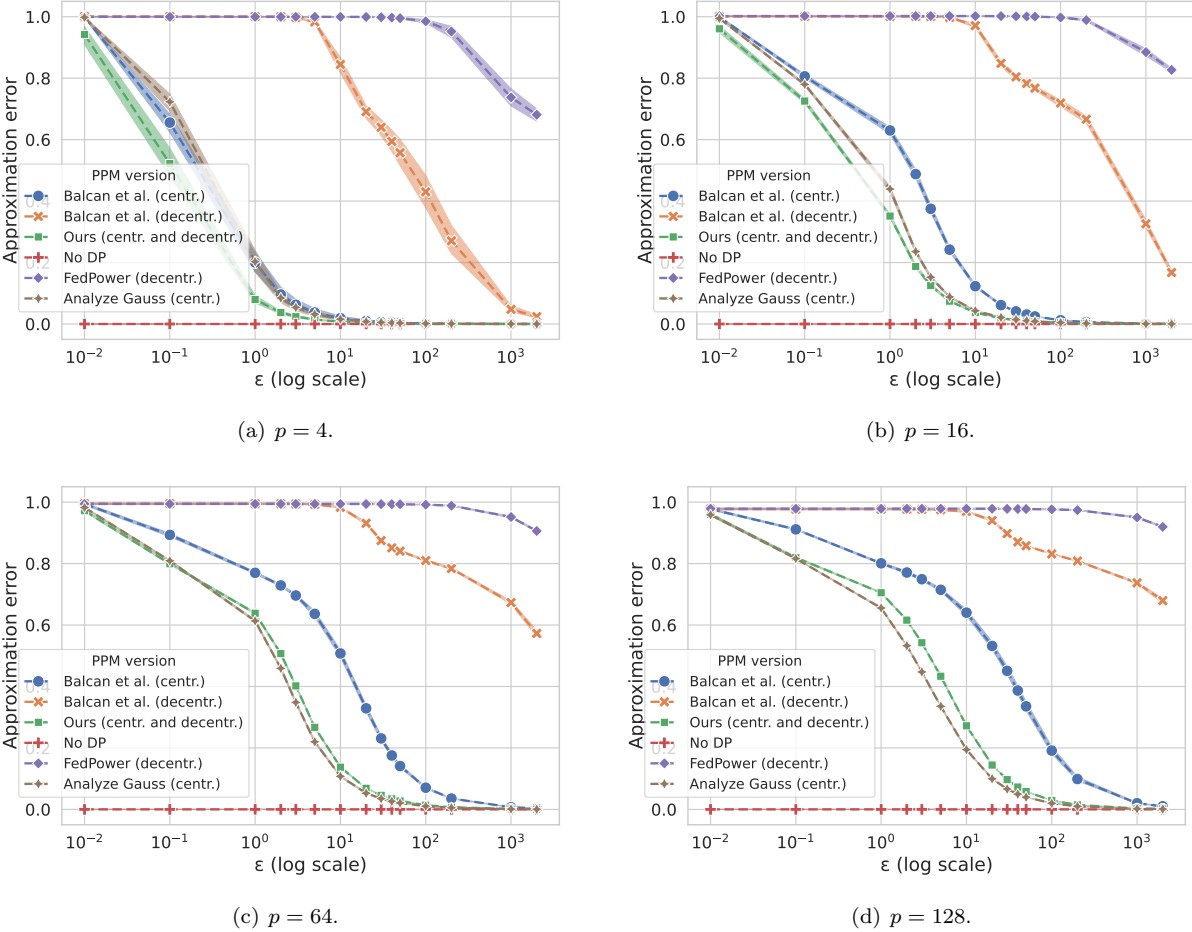

(a) $p = 4$.

(b) $p = 16$.

(c) $p = 64$.

(d) $p = 128$.

Figure 4: Impact of the Differential Privacy parameter $\epsilon$ on the relative approximation error $\frac{\|\tilde{\boldsymbol{R}}_p - \boldsymbol{R}_p\|_F}{\|\boldsymbol{R}_p\|_F}$, on the EachMovie dataset for varying $p = 4, 16, 64, 128$, $L = 3$, and $\delta = 10^{-4}$. Results are means of 10 runs with 99% bootstrap confidence intervals.

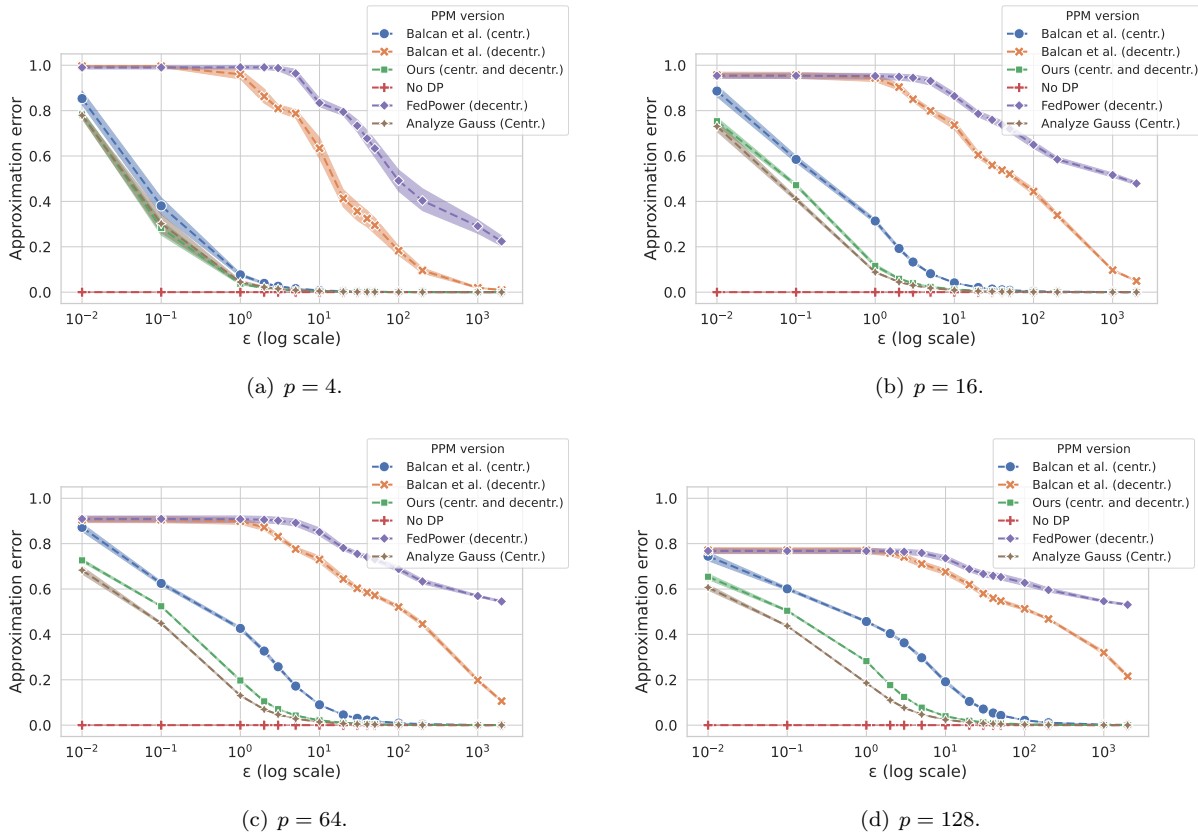

(a) $p = 4$.

(b) $p = 16$.

(c) $p = 64$.

(d) $p = 128$.

Figure 5: Impact of the Differential Privacy parameter $\epsilon$ on the relative approximation error $\frac{\|\tilde{\boldsymbol{R}}_p - \boldsymbol{R}_p\|_F}{\|\boldsymbol{R}_p\|_F}$, on the Jester dataset for varying $p = 4, 16, 32, 64$, $L = 3$, and $\delta = 10^{-4}$. Results are means of 10 runs with 99% bootstrap confidence intervals.

## C    Matrix-agnostic comparison of the runtime-dependent bounds

We provide here the derivation for Approximation 6.2:

*Proof.* Let $\boldsymbol{M} \in \mathbb{R}^{n \times k}$ with i.i.d $\mathcal{N}(0,1)$-distributed entries. Let $\boldsymbol{M} = \boldsymbol{Q}\boldsymbol{R}$ be its QR factorization (by definition $\boldsymbol{Q}$ is orthogonal and $\boldsymbol{R}$ upper triangular). By the Barlett decomposition theorem (Muirhead, 2009), we know that $\boldsymbol{Q}$ is a random matrix distributed uniformly in the Stiefel manifold $\mathbb{V}_{k,n}$. Then by Theorem 2.2.1 of Chikuse (2012), we know that a random matrix $\boldsymbol{Q}$ uniformly distributed on $\mathbb{V}_{k,n}$ can be expressed as $\boldsymbol{Q} = \boldsymbol{Z}(\boldsymbol{Z}^\top \boldsymbol{Z})^{-\frac{1}{2}}$ with $\boldsymbol{Z}$ another matrix with i.i.d $\mathcal{N}(0,1)$-distributed entries. We approximate $\boldsymbol{Z}^\top \boldsymbol{Z}$ as a diagonal matrix and remark that its diagonal elements are distributed as random chi-squared variables with $n$ degrees of liberty. For a matrix $\boldsymbol{A}$, we denote by $\boldsymbol{A}^{\circ n}$ the elementwise Hadamard exponentiation. Then we have:

$$\boldsymbol{Q}_{ij}^2 = (\boldsymbol{Z}(\boldsymbol{Z}^\top \boldsymbol{Z})^{-\frac{1}{2}})_{ij}^2 \tag{77}$$

$$= (\boldsymbol{Z}^{\circ 2}((\boldsymbol{Z}^\top \boldsymbol{Z})^{-\frac{1}{2}})^{\circ 2})_{ij} \tag{78}$$

$$= (\boldsymbol{Z}^{\circ 2}(\boldsymbol{Z}^\top \boldsymbol{Z})^{-1})_{ij}. \tag{79}$$

We remark that each element of $\boldsymbol{Z}^{\circ 2}$ is distributed as a chi-squared variable with one degree of freedom, and therefore $n \cdot \boldsymbol{Q}_{ij}^2$ is distributed as an i.i.d $F(1,n)$ random variable by definition. For large $n$, we can approximate these $F$ variables as Gaussians with mean $\mu = \frac{n}{n-2}$ and variances $\sigma^2 = \frac{2n^2(n-1)}{(n-2)^2(n-4)}$. Therefore

$\boldsymbol{Q}_{ij}^2 \sim \mathcal{N}(\frac{\mu}{n}, \frac{\sigma^2}{n^2})$ and:

$$k\boldsymbol{Q}_{ij}^2 \sim \mathcal{N}(\frac{k\mu}{n}, \frac{k^2\sigma^2}{n^2}), \qquad\qquad \sum_j \boldsymbol{Q}_{ij}^2 \sim \mathcal{N}(\frac{k\mu}{n}, \frac{k\sigma^2}{n^2}). \tag{80}$$

We can approximate the expectation of the maximum (noted as m) of $d$ Gaussian variables distributed as $\mathcal{N}(\mu_2, \sigma_2^2)$ by $m = \sigma_2\sqrt{2 \cdot \log(d)} + \mu_2$ by Lemma 2.3 from Massart (2007). We therefore get approximations of $\mathbb{E}[\max_{ij} k\boldsymbol{Q}_{ij}^2]$ and $\mathbb{E}[\max_i \|\boldsymbol{Q}_{i:}\|^2]$. We assume that the variances of $\Delta_l^{prior}$ and $\hat{\Delta}_l$ are small (because $n$ and $k$ are large) and therefore:

$$\mathbb{E}[\Delta_l^{prior}]^2 \approx \mathbb{E}[(\Delta_l^{prior})^2] = \sqrt{\frac{k^2\sigma^2}{n^2}}\sqrt{2 \cdot \log(kn)} + \frac{k\mu}{n}, \tag{81}$$

$$\mathbb{E}[\hat{\Delta}_l]^2 \approx \mathbb{E}[(\hat{\Delta}_l)^2] = \sqrt{\frac{k\sigma^2}{n^2}}\sqrt{2 \cdot \log(n)} + \frac{k\mu}{n}. \tag{82}$$

$\square$

