# OpenReview forum: "Differentially private and decentralized randomized power method"
_TMLR — Under review for TMLR_

### Review · Reviewer_HuEz · 2025-12-04

**Summary Of Contributions:**

Overview: This paper studies a differentially private version of the randomized power method, a method for approximating the top eigenspace of a matrix. The main contribution is a new bound on the sensitivity of a single step of the iterative algorithm. This tighter sensitivity bound allows the authors to achieve practically relevant improved performance. They demonstrate the improvement with both theoretical bounds, and empirical experiments on recommendation data.

The authors also discuss how secure aggregation can be incorporated into their algorithm to obtain a decentralized algorithm with the same performance as the central model algorithm (i.e. when the data is centrally located). Since the relevant part of the algorithm is linear with Gaussian noise addition, this extension is pretty standard.

Comments:
- The main technical contribution is the new sensitivity bound in Theorem 3.1. The proof of this new bound is fairly straightforward once one has the right adjacency relation. However, the observation that is improves over prior upper bounds on the sensitivity is nice. I thought Section 6.2 did a good job of highlighting the improvement and would have liked this discussion earlier. Without this discussion in Section 3, it was not clear that the new bound was a significant improvement (in particular it wasn’t clear if the sqrt(p) dependence was actually eliminated or was just folded into the notation).
- The comparison to Balcan et al. 2016 and Guo et al. 2024 as local model algorithms seems unfair. The addition of secure aggregation to make these algorithms into algorithms in the aggregate model of differential privacy is standard (again since the relevant part of the algorithm is linear with Gaussian noise addition). In my opinion, they should be compared as aggregate model algorithms with the same privacy accounting. This would not greatly affect the author’s overall story since the new sensitivity bound would still mean the author’s algorithm outperforms prior work.
- The most impressive part of the empirical results is that the performance of the author’s algorithm is on par with AnalyzeGauss. This is a central model algorithm with high compute since it involves computing the noising the full covariance matrix and computing its exact SVD decomposition. The proposed algorithm is significantly lighter weight and in a more realistic privacy model. This parity with AnalyzeGauss is not achieved in prior work that the author’s compare to.

Minor comments:
- The statement that DP mechanisms are always defined by the sensitivity of the function they are trying to compute is an oversimplification (even of the algorithm that is the central focus of this paper).
- The notation X^L looks like a power. I don’t have a solution to this, but I was a bit confused initially.
- I would have appreciated more discussion after each convergence theorem about what had changed. I found myself flicking back and forth between pages of the paper trying to identify the difference. Having this spelled out after Theorems 2.2 and 3.3 would make this simpler for the reader.
- The communication complexity analysis in Table 1 doesn’t seem to take into account any increase in communication complexity due to secure aggregation. My understanding is that this increase is not significant (constant factors?) but it should be mentioned.

**Audience:**

Yes

**Audience Explanation:**

The problem is an important one and the authors results make practically relevant improvements.

**Broader Impact Concerns:**

I have no ethical or broader impact concerns.

**Claims And Evidence:**

Yes

**Claims Explanation:**

I did not see any errors in the proof and did not notice any missing proofs.

**Requested Changes:**

- Clarify that Balcan et al. and Guo et al. can also be extended to algorithms in the aggregate model. Include these aggregate versions in the experiments as comparison points.
- Add discussion after Theorem 2.2 and 3.3 to aid readers in understanding the improvements.

---

> ### Author Response · Authors · 2026-06-25
> **Response**
>
> We thank the reviewer for the thoughtful and constructive comments. We
> particularly appreciate the suggestion to separate the contribution of
> Secure Aggregation from the contribution of the new sensitivity
> analysis, and to make the comparison to aggregate-model baselines more
> explicit.
>
> > **Reviewer comment.**
> >
> > “The main technical contribution is the new sensitivity bound in Theorem 3.1. The proof of this new bound is fairly straightforward once one has the right adjacency relation. However, the observation that is improves over prior upper bounds on the sensitivity is nice. I thought Section 6.2 did a good job of highlighting the improvement and would have liked this discussion earlier. Without this discussion in Section 3, it was not clear that the new bound was a significant improvement (in particular it wasn’t clear if the sqrt(p) dependence was actually eliminated or was just folded into the notation).”
>
> **Response.** We agree. The revised manuscript will explain the
> improvement immediately after the sensitivity theorem, not only in the
> empirical section. The previous calibration uses
> $\sqrt{p}\lVert \mathbf{X} \rVert_{\max}$, whereas our calibration uses
> $\max_i\|\mathbf{X}_{i:}\|_2$. The factor $\sqrt{p}$ is therefore not hidden
> in the notation; the new bound depends on the row $\ell_2$-norm of the
> iterate. As noted in response to Reviewer aZ8t, the two bounds can
> coincide in special cases, but the new quantity is no larger and is
> often much smaller in the regimes of interest.
>
> **Proposed revision.** We will add an explanatory paragraph immediately
> after Theorem 3.1 comparing $\sqrt{p}\lVert \mathbf{X} \rVert_{\max}$ with
> $\max_i\|\mathbf{X}_{i:}\|_2$. We will add interpretation paragraphs after
> Theorems 2.2 and 3.3 to identify precisely which dependence changes.
>
> > **Reviewer comment.**
> >
> > “The comparison to Balcan et al. 2016 and Guo et al. 2024 as local model algorithms seems unfair. The addition of secure aggregation to make these algorithms into algorithms in the aggregate model of differential privacy is standard (again since the relevant part of the algorithm is linear with Gaussian noise addition). In my opinion, they should be compared as aggregate model algorithms with the same privacy accounting. This would not greatly affect the author’s overall story since the new sensitivity bound would still mean the author’s algorithm outperforms prior work.”
>
> **Response.** We agree with this criticism. The original comparison
> mixed two effects: the privacy/communication model and the sensitivity
> calibration. To make the comparison fairer and more informative, we ran
> additional experiments on [EachMovie (link)](https://anonymous.4open.science/r/tmlrfigures-4883/eachmovie_secagg_local.pdf) and [MovieLens (link)](https://anonymous.4open.science/r/tmlrfigures-4883/movielens_secagg_local.pdf)
> that distinguish the local-DP versions from the
> Secure-Aggregation/aggregate-model versions of the different methods.
>
> **Proposed revision.** We will add the experiments on EachMovie and
> MovieLens comparing local-DP versions and SecAgg/aggregate-model
> versions of the private power-method baselines.
>
> > **Reviewer comment.**
> >
> > “The notation X^L looks like a power. I don’t have a solution to this, but I was a bit confused initially.”
>
> **Proposed revision.** To reduce ambiguity, we will revise the notation
> to use subscripts for iteration indices where possible, using
> $\mathbf{X}_\ell$ and $\mathbf{X}_L$ for iteration indices.
>
> > **Reviewer comment.**
> >
> > “I would have appreciated more discussion after each convergence theorem about what had changed. I found myself flicking back and forth between pages of the paper trying to identify the difference. Having this spelled out after Theorems 2.2 and 3.3 would make this simpler for the reader.”
>
> **Proposed revision.** We will add theorem-interpretation paragraphs
> after Theorems 2.2 and 3.3. These paragraphs will explicitly compare the
> old and new dependencies, explain how the improved sensitivity bound
> reduces the DP noise at each iteration, and clarify which parts of the
> convergence proof are inherited from prior noisy power-method analyses
> and which parts change due to the new calibration.
>
> > **Reviewer comment.**
> >
> > “The communication complexity analysis in Table 1 doesn’t seem to take into account any increase in communication complexity due to secure aggregation. My understanding is that this increase is not significant (constant factors?) but it should be mentioned.”
>
> **Response.** Table 1 already contains an additional overhead term for
> the proposed method, but this should be explained more clearly.
>
> **Proposed revision.** We will revise the caption and surrounding text
> to state which communication terms come from the randomized power-method
> updates and which come from Secure Aggregation. We will also mention
> that exact constants depend on the chosen Secure-Aggregation protocol,
> dropout handling, and quantization scheme.

---

### Review · Reviewer_Nhzc · 2026-02-01

**Summary Of Contributions:**

The focus of this paper are DP randomized power methods for centralized and federated settings. Specifically, the authors address :

1. The dependence the noise on the iteration rank (p) due to a loose sensitivity analysis in the private setting: they eliminate the factor of $\sqrt{p}$ found in previous works (Hardt and Price, 2014; Balcan et al., 2016).
2. The impracticality of federated settings due to local DP noise addition and/or materializing full covariance matrices.

The technical contributions can be summarized as follows:
1. Generalized Adjacency: Previous work allowed only for element wise updates for symmetric matrices, resulting in a very restrictive setting where only changes in the diagonal are permitted, to preserve symmetry. The proposed adjacency is a generalization of previous definitions while allowing for symmetric updates to the target covariance matrix $A$.

2. Tighter Sensitivity Analysis: authors directly analyze the p-dimensional power method iterates(where $p>k$ is the oversampling rank) rather than extrapolating from the one-dimensional case, and derive a tighter sensitivity bound that eliminates the
$\sqrt{p}$ factor found in prior work (Hardt and Price, 2014; Balcan et al., 2016).

3. Decentralized Algorithm: The paper proposes a federated DP approach using Secure Aggregation. Under an “honest-but-curious” server assumption, this allows emulating the centralized DP utility, while avoiding the high noise penalties of Local DP or the high communication costs of full covariance methods like Analyze Gauss.

Overall, the paper is well-written and clear. The work is incremental, but tightening constants has a large practical value for privacy applications where the independence on p can significantly increase utility. This constitutes a major contribution to the field of privacy-preserving spectral analysis.

**Audience:**

Yes

**Audience Explanation:**

Spectral decomposition is a stepping stone of many machine learning tasks, and having practical algorithms for the private setting is key for deploying these pipelines and enabling real world applications in domains like healthcare. This paper bridges this gap by improving the approximation level of the power method. The sensitivity analysis results appear to be new and practical for the audience. The results are supported by empirical results on publicly available realistic datasets.

**Broader Impact Concerns:**

No broader impact concerns, this is a theoretical paper using publicly available datasets.

**Claims And Evidence:**

Yes

**Claims Explanation:**

In brief, the claims are generally well-supported by theoretical proofs and numerical experiments, although some details regarding the decentralized protocols require clarification and some experiments in the appendix could make the main body more convincing (see following sections of the review).

1. Theoretical Analysis: The derivation of the improved sensitivity bound $\Delta$ under the more general adjacency notion is supported by proofs and its practical implications supported in realistic benchmarks.

2. Empirical Evidence: The experiments on recommender system datasets (MovieLens, EachMovie, etc.) demonstrate that the proposed method achieves lower approximation errors than decentralized and centralized baselines at comparable privacy budgets (epsilon). Even for Analyze Gauss, that demonstrates similar or superior empirical performance in some settings the paper explains the advantage based on communication/memory practicality rather than raw utility. The authors provide code to reproduce their experiments.

**Requested Changes:**

The paper is clearly organized and easy to follow. Some changes that I believe could strengthen the paper, in particular for people wanting to directly use the methods, are the following.

1. The authors claim that, assuming $A=RR^\top$, the adjacency notion allows for element-wise updates on the original $R$ matrix. The discussion following eq. 7 suggests that one can change one user-item interaction but not full rows (one full user), which is more common in privacy applications, where user-level DP and not record level DP is required. I think the sensitivity analysis is still valid for user level updates with more than one element change in R, by replacing the bound of 1 in

$\sqrt{\sum \| C_i \|}\leq 1$ by some constant. This constant could be  application dependent, e.g. by adding contribution bounding, such as one user can contribute at most 10 items... Is this true or am I misunderstanding something?

2. The claim of a "decentralized" method is slightly ambiguous. Algorithm 2 explicitly relies on a "central node" for broadcasting and computing the secure aggregation. While this does not require the central node to see raw data, it suggests a federated setting with a coordinator rather than a peer-to-peer decentralized setting. The distinction between "federated with SecAgg" and "fully decentralized" could be made more precise.

3. The paper mentions using SecAgg and cites Bonawitz et al., but Algorithm 2 treats SecAgg as a black box. Given that the authors claim this as one of the main contributions, I think that making more clear the limitations of SecAgg's specific protocols would make the paper more transparent in this front (I acknowledge that this is briefly discussed in the conclusion). When first introduced I suggest adding more details on the specific protocols, rather than directing readers to the references. In particular, the reader may directly disregard this contribution due to assumed computational cost and/or the limit of users that can contribute per round. (I may be outdated in more recent secagg algorithms, but precisely adding briefly the practical overhead of the chosen protocol would strengthen the "low overhead" claim).

4. Figure 1 clearly shows the advantage of the improved bound for various values of $\epsilon$. Given that the contribution of the paper is removing the dependence on $p$, I would suggest having  the approximation error vs $p$ instead. This is partially discussed in Appendix B.3, so moving Figure 4 or 5 (or a summary) into the main paper to substantiate the claims regarding improved scaling with target rank would make the claims more evident.

Minor clarifications:

1. It is unclear how the columns in Table 2 are computed. Is $p$ set to $m$?  please add a caption explaining how $\Delta_l$ was derived.

2. I would add to the caption of Figure 1 the reason why Analyze Gauss is excluded in the left panel since it is confusing before getting to  the paragraph where the cost for large-scale datasets prohibits its application to this dataset.

---

> ### Author Response · Authors · 2026-06-25
> **Response**
>
> We thank the reviewer for the detailed and constructive feedback. The comments help clarify the privacy granularity, the meaning of decentralization in our protocol, and the assumptions behind Secure Aggregation.
>
> > **Reviewer comment.**
> >
> > “The authors claim that [...] the adjacency notion allows for element-wise updates on the original $R$ matrix. [...] Is the sensitivity analysis still valid for user-level updates with contribution bounding, e.g., limiting each user to 10 items?”
>
> **Response.** This is an important clarification. Our current results provide record-level protection for bounded changes corresponding to individual interactions, not unrestricted full-row user-level DP. The general adjacency relation was introduced because a change in the interaction matrix $\mathbf{R}$ can induce a structured symmetric update in the derived matrix $\mathbf{A}$ rather than a single-entry change in $\mathbf{A}$. We will make this distinction explicit and avoid implying that unrestricted user-level updates are covered without contribution bounding.
>
> **Proposed revision.** After the adjacency definition in Section 3, we will state that the main results concern record-level protection. We will then explain that contribution-bounded user-level privacy can be obtained by assuming
>
> $$
> \sqrt{\sum_i|\mathbf{C}_{i:}|_1^2}\leq B,
> $$
>
> which changes the sensitivity bound to
>
> $$
> \Delta_\ell \leq B\max_i|\mathbf{X}_{\ell,i:}|_2.
> $$
>
> Here, $B$ is determined by the clipping or contribution bound imposed on each user.
>
> > **Reviewer comment.**
> >
> > “The claim of a ‘decentralized’ method is slightly ambiguous. Algorithm 2 relies on a central node [...], suggesting a federated setting with a coordinator rather than a fully peer-to-peer setting.”
>
> **Response.** We agree. Algorithm 2 is more precisely a federated protocol with a coordinating server and Secure Aggregation, rather than a fully peer-to-peer protocol.
>
> **Proposed revision.** We will replace “decentralized” with “federated” or “aggregate-model” when referring to Algorithm 2. This terminology will be revised consistently in the title, abstract, introduction, Section 5 heading, Algorithm 2, and conclusion. We will also distinguish this coordinating-server setting from fully peer-to-peer secure-averaging protocols.
>
> > **Reviewer comment.**
> >
> > “Algorithm 2 treats SecAgg as a black box. [...] Adding the assumptions, practical limitations, and overhead of the chosen protocol would strengthen the ‘low overhead’ claim.”
>
> **Response.** We agree. The manuscript abstracts Secure Aggregation to keep the algorithm readable, but its assumptions, limitations, and practical overhead should be stated more clearly.
>
> **Proposed revision.** At the beginning of Section 5, we will state that the simplified analysis assumes honest-but-curious participants, no client dropout, and omits finite-field quantization and modular-arithmetic errors. We will explain that practical Secure-Aggregation implementations additionally require dropout recovery and finite-precision encoding, which introduce communication and implementation overhead beyond the simplified model.
>
> > **Reviewer comment.**
> >
> > “Given that the contribution is removing the dependence on $p$, I would suggest showing approximation error versus $p$. [...] Moving Figure 4 or 5, or a summary, into the main paper would make this claim more evident.”
>
> **Response.** We agree. The approximation-error-versus-$p$ evidence is directly connected to the main sensitivity improvement and should be easier to find.
>
> **Proposed revision.** In Section 6, we will summarize this experiment and explicitly connect it to the sensitivity bound tightening. The complete plots and experimental details will remain in Appendix B.3, with a direct pointer from the main text.
>
> > **Reviewer comment.**
> >
> > “It is unclear how the columns in Table 2 are computed. Is $p$ set to $m$? Please explain how $\Delta_\ell$ was derived.”
>
> **Response.** Thank you for pointing this out. Table 2 reports sensitivity quantities obtained from the coherence-based runtime-independent bounds. The dataset-specific constants are derived from the estimated coherence terms and number of items, while the dependence on $p$ remains explicit.
>
> **Proposed revision.** We will revise the caption and surrounding text to state exactly how $\Delta_{\ell}^{prior}$ and $\widehat{\Delta}_{\ell}$ are computed. We will also clarify before Table 2 that $p$ is the iteration rank and $n$ is the number of items in each dataset.
>
> > **Reviewer comment.**
> >
> > “Please explain in the Figure 1 caption why AnalyzeGauss is excluded from the left panel.”
>
> **Proposed revision.** We will add to the caption: “AnalyzeGauss is omitted from the MovieLens-10M panel due to its prohibitively high memory requirements.”

---

### Review · Reviewer_aZ8t · 2026-06-18

**Summary Of Contributions:**

The paper studies differentially private variants of the randomized power method, an important tool for problems such as $k$-PCA. It makes two main contributions. First, it provides a new privacy analysis under a more general notion of adjacency. Second, it develops a decentralized variant of the algorithm.

For the first contribution, the underlying algorithm is known and has appeared in prior work. Existing analyses establish privacy guarantees under an entry-level adjacency relation: two matrices $A, A' \in \mathbb{R}^{n \times n}$ are adjacent if they differ in at most one entry, and the magnitude of this difference is at most $1$. More precisely, there exist indices $i,j$ such that $|A_{ij}-A'_{ij}| \leq 1$, while all other entries of $A$ and $A'$ are identical.

The present paper considers the more general adjacency relation
$\sqrt{\sum_{i=1}^n \lVert (A-A')\_{i:} \rVert\_1^2} \leq 1$
and provides a new privacy analysis of the private randomized power method under this model. The main technical difference lies in the sensitivity bound used in the analysis. The randomized power method is iterative, and at each iteration it maintains a matrix $X \in \mathbb{R}^{n \times p}$. Prior work bounds the relevant sensitivity by $\sqrt{p}\lVert X\rVert\_{\max}$, whereas the present paper uses the bound $\max_i \lVert X_{i:}\rVert\_2$. The latter can be substantially tighter in a broad range of parameter regimes, and the paper also provides experimental evidence supporting this improvement.

The second contribution is a decentralized variant of the private randomized power method.

The key contribution of the paper is new privacy analysis, as essentially everything follows from that. I consider this to be a nice result that is worth publishing.

**Audience:**

Yes

**Audience Explanation:**

This is a paper on privacy and federated learning, which I think are topics that are of interest to the TMLR community.

**Broader Impact Concerns:**

I do not consider the contents of the paper to lead to broader ethical concerns.

**Claims And Evidence:**

Yes

**Claims Explanation:**

The main proofs are morally correct, as far as I could tell.

**Requested Changes:**

I feel one claim needs to be slightly toned down. The new sensitivity bound, while stronger than the older one, is not uniformly better, in the sense that the two bounds would give the same result for a matrix $X$ that has a row whose elements are all equal to the largest element of $X$. While this may be a low-probability event, I think a comment mentioning when the two bounds give the same results should be included.

Additionally, the experimental section comes across as somewhat restricted, since all the experiments are conducted under the assumptions that $p = 32$ and $L = 3$. Also, it is not clarified what the employed value for $\delta$ is. I feel these are issues that the authors should address, both clearly stating what $\delta$ they use, and by working with different combinations of $p$ and $L$.

---

> ### Author Response · Authors · 2026-06-25
> **Response**
>
> We thank the reviewer for the careful and encouraging assessment of the
> paper. We are glad that the reviewer found the privacy analysis to be a
> useful contribution and the main proofs to be sound. We also appreciate
> the requests to state the sensitivity improvement more precisely and to
> make the experimental settings clearer.
>
> > **Reviewer comment.**
> >
> > “I feel one claim needs to be slightly toned down. The new sensitivity bound, while stronger than the older one, is not uniformly better, in the sense that the two bounds would give the same result for a matrix $X$ that has a row whose elements are all equal to the largest element of $X$. While this may be a low-probability event, I think a comment mentioning when the two bounds give the same results should be included.”
>
> **Response.** We agree. The revised paper will avoid wording that could
> suggest a strict improvement in every possible case. We will make this
> point explicit so that the claim is accurate and not overstated.
>
> **Proposed revision.** We will revise the abstract and introduction to
> describe the proposed sensitivity bound as always no larger than, rather
> than strictly smaller than, the previous bound. Immediately after
> Theorem 3.1, we will state that equality can occur when one row contains
> $p$ entries whose magnitudes are all equal to $\|\mathbf{X}\|_{\max}$.
>
> > **Reviewer comment.**
> >
> > “Additionally, the experimental section comes across as somewhat restricted, since all the experiments are conducted under the assumptions that $p = 32$ and $L = 3$. Also, it is not clarified what the employed value for $\delta$ is. I feel these are issues that the authors should address, both clearly stating what $\delta$ they use, and by working with different combinations of $p$ and $L$.”
>
> **Response.** We agree that the experimental settings should be made
> more explicit and that the empirical section should better illustrate
> the dependence on the main algorithmic parameters.
>
> 1.  On $\delta$: We use $\delta=10^{-4}$ in the experiments, chosen with
>     respect to the number of items: MovieLens-10M contains $10{,}677$
>     items, so this value is approximately $1/10{,}677$. We keep the same
>     $\delta$ across all datasets and methods for a consistent noise
>     scale comparison. Since the noise variance $\sigma^2$ depends on
>     $\epsilon$ and $\delta$, fixing $\delta$ and varying $\epsilon$
>     allows us to compare the methods across a range of noise variance
>     values. We acknowledge that practical applications could choose
>     other values of $\delta$.
>
> 2.  On $p$: The ablation study on $p$ is already included in the
>     appendix, but we will add a clearer pointer to it from the main
>     text.
>
> 3.  On $L$: In addition, we ran a new ablation on the number of power
>     iterations $L$:
>
>     [L=1](https://anonymous.4open.science/r/tmlrfigures-4883/iterations_filtered_relative_froberror_comparison_eachmovie_new_components32_iterations1.pdf);
>     [L=2](https://anonymous.4open.science/r/tmlrfigures-4883/iterations_filtered_relative_froberror_comparison_eachmovie_new_components32_iterations2.pdf);
>     [L=4](https://anonymous.4open.science/r/tmlrfigures-4883/iterations_filtered_relative_froberror_comparison_eachmovie_new_components32_iterations4.pdf);
>     [L=5](https://anonymous.4open.science/r/tmlrfigures-4883/iterations_filtered_relative_froberror_comparison_eachmovie_new_components32_iterations5.pdf).
>
> **Proposed revision.** In the experimental setup of Section 6, we will
> add: “We use $\delta=10^{-4}$ in all experiments. This value was
> selected relative to the $10{,}677$ items in MovieLens-10M, so that
> $\delta\approx 1/10{,}677$, and is kept fixed across all datasets and
> methods. Because the Gaussian noise variance depends jointly on
> $\epsilon$ and $\delta$, varying $\epsilon$ at fixed $\delta$ evaluates
> the methods over a range of noise levels. Other $(\epsilon,\delta)$
> pairs may induce the same or comparable noise levels, and different
> values of $\delta$ may be appropriate in practical applications.” We
> will also add an explicit pointer from Section 6 to the existing
> $p$-ablation in Appendix B.3 and add the new experiment reporting
> approximation error for several values of $L$.